# Targeting EGFR-dependent tumors by disrupting an ARF6-mediated sorting system

Huiling Guo[1,6], Juan Wang[2,6], Su Ren[1,6], Lang-Fan Zheng[2,6], Yi-Xuan Zhuang[1], Dong-Lin Li[1], Hui-Hui Sun[1], Li-Ying Liu[1], Changchuan Xie[1], Ya-Ying Wu[1], Hong-Rui Wang[1], Xianming Deng[1,3], Peng Li[2,4,5] & Tong-Jin Zhao[2,5] ✉

Aberrant activation of EGFR due to overexpression or mutation is associated with poor prognosis in many types of tumors. Here we show that blocking the sorting system that directs EGFR to plasma membrane is a potent strategy to treat EGFR-dependent tumors. We find that EGFR palmitoylation by DHHC13 is critical for its plasma membrane localization and identify ARF6 as a key factor in this process. N-myristoylated ARF6 recognizes palmitoylated EGFR via lipid-lipid interaction, recruits the exocyst complex to promote EGFR budding from Golgi, and facilitates EGFR transporting to plasma membrane in a GTP-bound form. To evaluate the therapeutic potential of this sorting system, we design a cell-permeable peptide, N-myristoylated GKVL-TAT, and find it effectively disrupts plasma membrane localization of EGFR and significantly inhibits progression of EGFR-dependent tumors. Our findings shed lights on the underlying mechanism of how palmitoylation directs protein sorting and provide an potential strategy to manage EGFR-dependent tumors.

Epidermal growth factor receptor (EGFR) is a receptor tyrosine kinase that regulates epithelial development and homeostasis[1]. Aberrant activation of EGFR due to overexpression or mutation is a driving force of many types of tumors, including lung and breast cancer[1,2]. Currently, EGFR-specific tyrosine kinase inhibitors (TKIs) and monoclonal antibodies are potent strategies to target EGFR mutant cancers though with acquired resistance, but they did not work well in EGFR-overexpression tumors[1,3–5]. New strategies need to be developed to target tumors with EGFR overexpression or acquired mutations.

Sorting proteins to their destinations based on the intrinsic sorting signals is a prerequisite for their physiological functions[6,7]. Protein S-palmitoylation, hereafter called palmitoylation, is a post-translational lipid modification that the palmitoyl group is attached to the thiol group of a cysteine residue through thioester bond catalyzed by a family of enzymes containing the Asp-His-His-Cys (DHHC) motif[8,9]. It has been increasingly appreciated that

palmitoylation can function as a sorting signal to direct proteins to specific membranes[10–12]; however, the underlying mechanism remains unknown.

EGFR has been reported to be a palmitoylated protein, although the palmitoylation sites and the enzymes vary in different reports[13–15]. It remains unclear about whether palmitoylation is required for targeting EGFR to the plasma membrane (PM). In one report, palmitoylation of EGFR is required for its PM localization and activation[13], whereas in the other report, palmitoylation of EGFR is not required for its PM localization[14]. Here, to address the issue, we show that either inhibition of the palmitoylation of EGFR or mutation of its palmitoylation sites abolished the PM localization of EGFR, indicating that palmitoylation is essential to target EGFR to PM. Furthermore, we identified ARF6 as a key small GTPase in targeting palmitoylated EGFR from Golgi to PM. ARF6 recognizes palmitoylated EGFR with its N-myristoylation, recruits the exocyst complex with Lys3 to promote EGFR budding from Golgi, and gets

[1]State Key Laboratory of Cellular Stress Biology, School of Life Sciences, Xiamen University, Xiamen, Fujian 361102, China. [2]State Key Laboratory of Genetic Engineering, Shanghai Key Laboratory of Metabolic Remodeling and Health, Institute of Metabolism and Integrative Biology, Zhongshan Hospital, Fudan University, Shanghai 200438, China. [3]State-province Joint Engineering Laboratory of Targeted Drugs from Natural Products, Xiamen University, Xiamen, Fujian 361102, China. [4]School of life sciences, Zhengzhou University, Zhengzhou, Henan 450001, China. [5]Shanghai Qi Zhi Institute, Shanghai 200232, China. [6]These authors contributed equally: Huiling Guo, Juan Wang, Su Ren, Lang-Fan Zheng. ✉e-mail: zhaotj@fudan.edu.cn

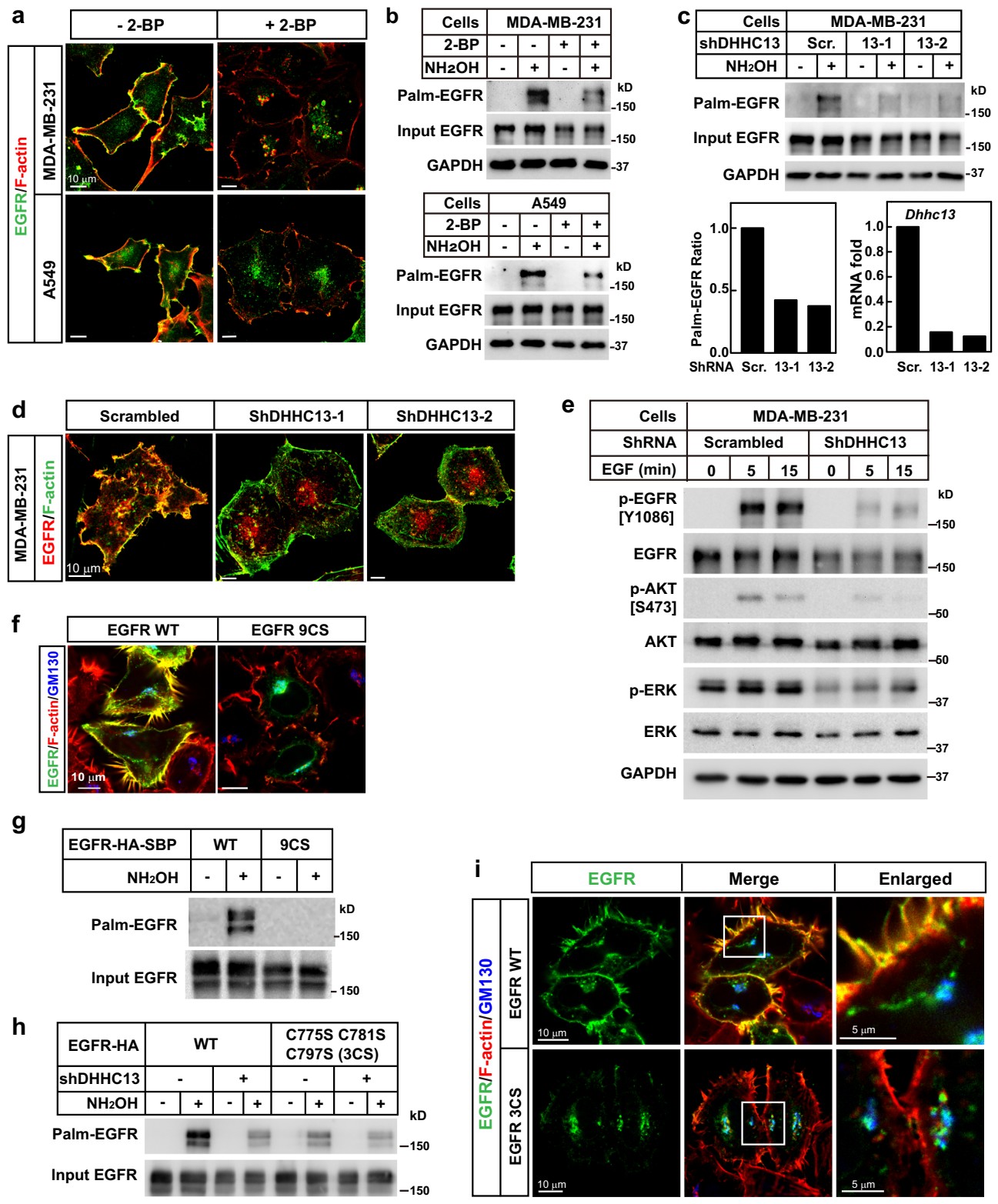

converted to GTP-bound form by EFA6B to facilitate vesicle trafficking to PM. We further explored the therapeutic value of our findings and designed a cell-permeable peptide, myristoylated GKVL-TAT, and found that it could effectively inhibit the growth of EGFR-overexpression tumors. Our findings have thus not only uncovered a sorting mechanism for palmitoylated proteins from Golgi to PM, but also proposed a potential strategy to treat EGFR-overexpression tumors.

## Results

### Palmitoylation of EGFR by DHHC13 is required for its PM localization

To explore the effect of palmitoylation on the sorting of EGFR to PM, we first treated two cell lines expressing endogenous EGFR, a triple-negative breast cancer (TNBC) cell line MDA-MB-231, and a non-small cell lung cancer (NSCLC) cell line A549[16,17], with 2-bromopalmitate (2-BP), an inhibitor of protein palmitoylation. As shown in Fig. 1a, b, 2-BP

**Fig. 1 | EGFR Palmitoylation by DHHC13 is required for its PM localization.**
**a**, **b** On day 0, MDA-MB-231 cells or A549 cells were set up at $4 \times 10^4$ cells per 35-mm dish. On day 2, cells were switched to a serum-free medium and incubated with or without 100 µM 2-BP overnight, harvested for immunostaining with an anti-EGFR antibody (**a**) and acyl-RAC assay (**b**). Rhodamine-labeled phalloidin was used in (**a**) to stain F-actin to indicate PM. **c**, **d** On day 0, MDA-MB-231 cells were infected with lentivirus expressing scrambled shRNA or shDHHC13. On day 2, cells were selected with 2 µg/ml puromycin. On day 4, cells were subjected to Acyl-RAC assay (**c**) or immunostaining using anti-EGFR antibody and FITC-phalloidin (for F-actin) (**d**). The intensities of indicated bands were quantified using the VisionWorks software on a ChemStudio imaging system and the ratios of palm-EGFR were shown in the left panel. The knockdown efficiency of *DHHC13* was shown in the right panel. **e** Control and DHHC13 knockdown MDA-MB-231

cells were serum-starved overnight, treated with 100 ng/ml EGF and harvested at the indicated time for western blot. **f**, **g** On day 0, HeLa cells were set up $2 \times 10^4$ cells per 35-mm dish. On day 2, cells were transfected with WT or 9CS mutant of EGFR-Flag (0.5 µg each). On day 3, cells were harvested for immunostaining (**f**) or acyl-RAC assay (**g**). **h** On day 0, HEK293T cells transduced with scrambled or DHHC13 shRNA were transfected with WT or 3CS (C775/C781/C797S) mutant of EGFR-HA-SBP. On day 2, cells were harvested and subjected to an acyl-RAC assay. The input and eluted fractions were immunoblotted with anti-HA antibodies. **i** HeLa cells were set up at $2 \times 10^4$ cells per 35-mm dish. On day 1, cells were transfected with WT or 3CS mutant of EGFR-HA-SBP. On day 2, cells were switched to a serum-free medium for 4 h and fixed for immunostaining with anti-HA and anti-GM130 antibodies. In Panels **a**, **d**, **f**, **i**, the scale bar is 10 µm. Source data are provided as a Source Data file.

treatment decreased the palmitoylation of EGFR and disrupted the PM localization of EGFR. Treatment of HeLa cells expressing EGFR-Flag with 2-BP further confirmed the results (Supplementary Fig. 1a, b), suggesting that palmitoylation might be required for the PM localization of EGFR.

To confirm the results, we went on to identify the palmitoylating enzyme that is required for targeting EGFR to PM. As loss of palmitoylation trapped EGFR in Golgi (Supplementary Fig. 1a), we hypothesized that EGFR might be palmitoylated in Golgi before being sorted to PM. We, therefore, knocked down each of the nine DHHCs that showed Golgi localization[18], and found that the knockdown of DHHC13 or DHHC17 disrupted the PM localization of EGFR in MDA-MB-231 cells (Supplementary Fig. 1c, d). While the knockdown of DHHC13 dramatically decreased the palmitoylation of EGFR (Fig. 1c), the knockdown of DHHC17 did not show such an effect (Supplementary Fig. 1e). We thus focused on DHHC13, which was confirmed to show Golgi localization (Supplementary Fig. 1f). In both MDA-MB-231 and A549 cells, two separate shRNAs of DHHC13 disrupted the PM localization of EGFR (Fig. 1d and Supplementary Fig. 1g). Knockdown of DHHC13 did not affect the PM localization of DSC2 (Supplementary Fig. 1h), suggesting normal trafficking of other proteins. Furthermore, the loss of DHHC13 dramatically decreased the EGF-induced EGFR signaling pathway (Fig. 1e and Supplementary Fig. 1i). These data indicate that palmitoylation of EGFR by DHHC13 is required for its PM localization and function.

We then sought to know the palmitoylation sites that are required for EGFR PM localization. To clarify the palmitoylation sites of EGFR, we performed mass spectrometry and found that eight of the nine cytosolic of cysteines (Cys775, Cys797, Cys818, Cys939, Cys950, Cys1049, Cys1058, and Cys1146) were palmitoylated (Supplementary Fig. 2a and Supplementary Data 1). As only a few peptides containing Cys781 were detected, we could not rule out that Cys781 might also be palmitoylated. We thus mutated all nine Cys to Ser, and found that the mutant (9CS) was not palmitoylated and failed to localize on PM (Fig. 1f, g). We then moved on to identify the Cys residues that were palmitoylated by DHHC13. We generated EGFR mutants with eight of the nine Cys mutated to Ser and kept only one Cys. As shown in Supplementary Fig. 2b, all mutants were palmitoylated, indicating that all of the cytosolic cysteines of EGFR are palmitoylated. When DHHC13 was knocked down, palmitoylations of Cys775, Cys781, and Cys797, not the others, were decreased (Supplementary Fig. 2b). Mutation of these three Cys to Ser (3CS) dramatically decreased EGFR palmitoylation, and knockdown of DHHC13 only slightly decreased 3CS mutant palmitoylation (Fig. 1h), confirming that EGFR is the substrate of DHHC13. Similar to the 9CS mutant, the 3CS mutant was mainly trapped in the Golgi and failed to localize to the PM (Fig. 1i). And the 3CS mutant did not seem to affect the protein folding of EGFR, as mutating these sites in the cytosolic domain did not affect the profiling in a gel filtration analysis (Supplementary Fig. 2c). These data indicate that palmitoylation of EGFR at Cys775, Cys781, and Cys797 by DHHC13 is required for its PM localization.

## ARF6 is required for the sorting palmitoylated EGFR from Golgi to PM

To explore the role of palmitoylation in targeting EGFR to PM, we utilized the retention using selective hook (RUSH) system[19] in which a fusion protein of Golgin-84 and Streptavidin was used as a Golgi hook to synchronize the sorting of EGFR from Golgi (Fig. 2a). We co-expressed WT or 3CS mutant of EGFR-HA-SBP and Golgin-84-Streptavidin in HeLa Cells. Before biotin treatment, EGFR were trapped in Golgi and colocalized with GM130. After biotin treatment, WT EGFR reached PM, but the 3CS mutant did not (Fig. 2b), indicating that palmitoylation is required for sorting EGFR from Golgi to PM.

We next sought to know how palmitoylation directs EGFR from Golgi to PM. To identify the key factor in facilitating the sorting of palmitoylated EGFR, we expressed WT or 9CS mutant EGFR, isolated their interacting proteins, and found that ARF6, a small GTPase, specifically interacted with WT EGFR (Fig. 2c; Supplementary Fig. 3a; and Supplementary Data 1). We first knocked down ARF6 and found that two separate shRNAs could both disrupt the PM localization of EGFR (Supplementary Fig. 3b). To further confirm the results, we generated an $ARF6^{-/-}$ HeLa cell line using the CRISPR-Cas9 system (Supplementary Fig. 3c). We used RUSH systems to synchronize vesicle trafficking and studied EGFR sorting in WT and $ARF6^{-/-}$ cells. In WT cells, EGFR was released from Golgi and sorted to PM after biotin treatment; however, in $ARF6^{-/-}$ cells, biotin treatment did not release EGFR from Golgi (Fig. 2d). These results indicate that ARF6 is required for sorting EGFR from Golgi to PM.

We then utilized the RUSH system to study how ARF6 facilitates EGFR sorting from Golgi to PM. We co-expressed EGFR-HA-SBP, ARF6-Flag, and the Golgi hook in $ARF6^{-/-}$ cells. Before biotin treatment, EGFR, ARF6, and GM130 showed well colocalization, as illustrated by immunostaining and quantification of the fluorescent intensities (Fig. 2e). After biotin treatment, EGFR, together with ARF6, started to bud out of Golgi at 30 min, and by 1 h the majority of EGFR and ARF6 were colocalized on PM (Fig. 2e). To examine the role of palmitoylation of EGFR in the sorting process, we expressed EGFR 3CS. Figure 2f showed that EGFR 3CS was still trapped in Golgi after biotin treatment, indicating that palmitoylation is required for the budding of EGFR from Golgi. Notably, the 3CS mutant failed to recruit ARF6 to the Golgi, implying that palmitoylation of EGFR is required for its binding with ARF6 (Fig. 2f and Supplementary Fig. 3d, e).

We have also noticed that ARF proteins all have an N-terminal lipid modification, myristoylation[20]. To examine the role of N-myristoylation of ARF6, we generated the non-myristoylated G2A mutant of ARF6. Before biotin treatment, the G2A mutant did not show any enrichment in Golgi where EGFR was trapped (Fig. 2g). After biotin treatment, EGFR was still trapped in Golgi (Fig. 2g), indicating that the N-myristoylation of ARF6 is required for the budding of palmitoylated EGFR from Golgi.

To further verify the results, we performed proximity ligation assay[21]. As shown in Fig. 2h, i, while the fluorescent signals were clearly detected in cells expressing WT EGFR and WT ARF6, the intensity was

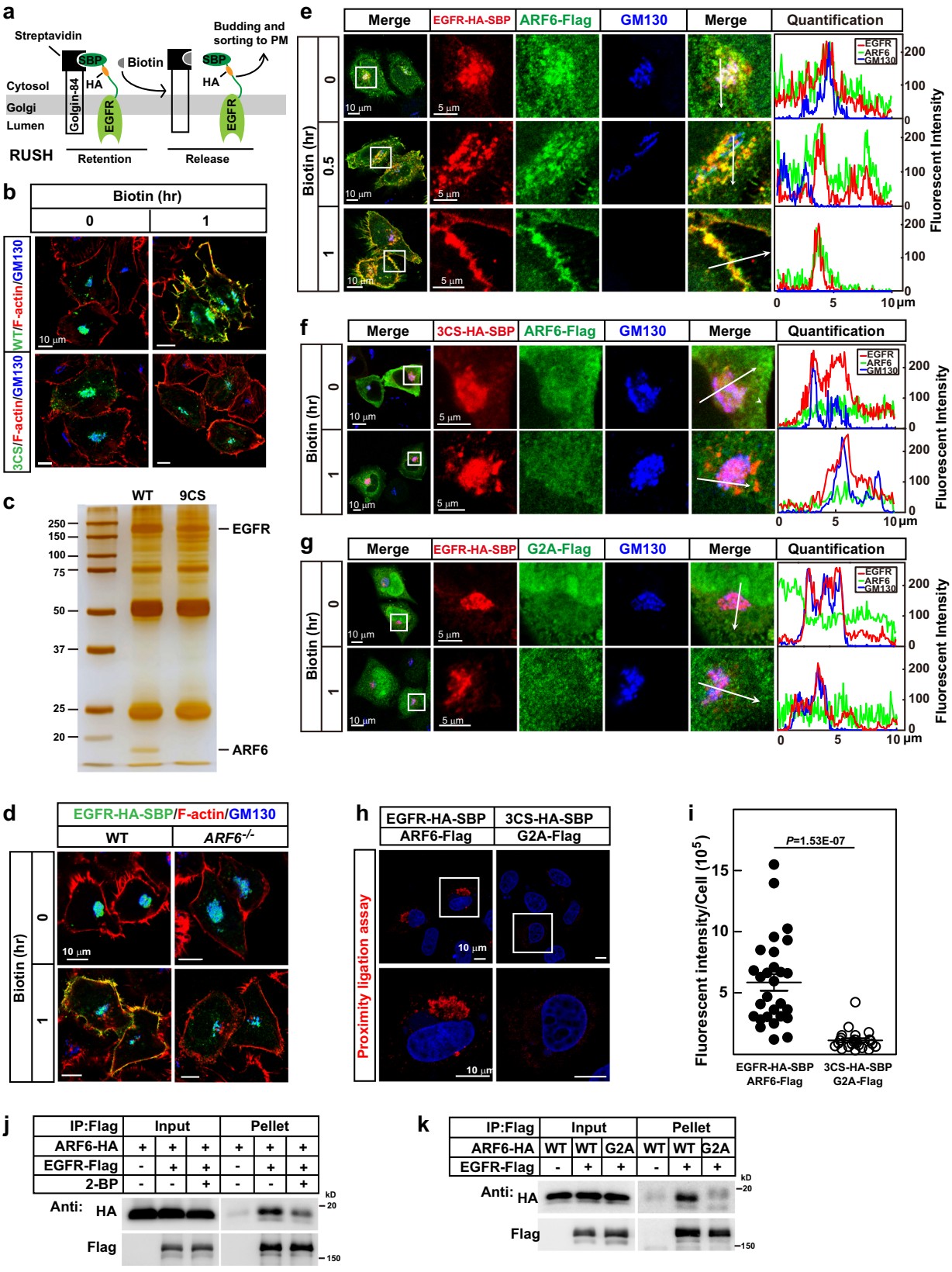

much lower in cells expressing the 3CS mutant of EGFR and G2A mutant of ARF6. Furthermore, we performed immunoprecipitation to evaluate the interaction between EGFR and ARF6, and found that eliminating either the palmitoylation of EGFR or myristoylation of ARF6 greatly decreased their interaction (Fig. 2j, k). These data further confirmed that ARF6 recognizes EGFR through lipid–lipid interaction.

## The N-terminus of ARF6 is required for EGFR budding from Golgi

As all of the mammalian ARFs have N-myristoylation[20], we asked why only ARF6 is required for EGFR sorting. Sequence alignment of the five human ARFs revealed that they were highly homologous, but ARF6 differed from the others in that it had a shorter N-terminus

**Fig. 2 | ARF6 is required for the sorting of EGFR to PM. a** A schematic illustration of the retention using a selective hook (RUSH) system to study the effect of ARF6 on sorting EGFR from Golgi to PM. **b** On day 0, HeLa cells were transfected with Streptavidin-Golgin-84 and WT or 3CS of EGFR-HA-SBP (0.5 µg each). On day 1, cells were fixed at 0 or 1 h after 100 µM D-biotin treatment and immunostained using anti-HA and anti-GM130 antibodies. **c** On day 0, WT or 9CS EGFR-HA-SBP was transfected with Streptavidin-Golgin-84 into HEK293T. On day 1, cells were washed with PBS and crosslinked with 2 mM DSP on ice for 2 h, which was stopped by 0.1 M Tris (pH 8.0). Immunoprecipitations were performed using anti-HA beads. The pellets were subjected to silver staining and mass spectrometry. **d** Twenty-four hours after transfection with EGFR-HA-SBP and Streptavidin-Golgin-84, WT and *ARF6*$^{-/-}$ HeLa cells were treated with 100 µM D-biotin and immunostained as in (**b**). **e–g** *ARF6*$^{-/-}$ HeLa cells were transfected with Streptavidin-Golgin-84 (**e–g**), WT (**e, g**) or 3CS (**f**) of EGFR-HA-SBP and WT (**e, f**)

or G2A (**g**) of ARF6-Flag (0.5 µg each). Cells were treated with 100 µM biotin and immunostained as in (**d**). Fluorescent intensity was quantified using ZEN2.3. Arrows indicate the quantification area. **h, i** HeLa cells were transfected with indicated plasmids with Streptavidin-Golgin-84. Proximity ligation assay was performed following the manufacturer's instructions. Intensities were quantified and plotted (**i**). Data were presented as mean ± SEM, EGFR-HA-SBP, and ARF6-Flag, *n* = 28; 3CS-HA-SBP and G2A-Flag, *n* = 24. A two-sided Student's *t*-test was performed. **j, k** HEK293T cells were transfected with indicated plasmids. Eight hours later, cells were incubated with 1 µM nocodazole overnight with or without 100 µM 2-BP, washed twice with PBS, and incubated in a fresh medium for another 0.5 h. Cells were treated with 2 mM DSP as in (**c**). Cell lysates were subjected to immunoprecipitation with anti-Flag M2 beads. Cells were pretreated with 2-BP for 24 h before harvest (**j**). Scale bars were as indicated. Source data are provided as a Source Data file.

(Supplementary Fig. 4a). To test whether the other four ARFs would have a similar function as ARF6, we performed a rescue experiment by introducing ARF1, ARF3, ARF4, or ARF5 into *ARF6*$^{-/-}$ cells, but none of the these ARFs would restore the PM localization of EGFR (Fig. 3a and Supplementary Fig. 4b). We then performed domain swapping by exchanging the N-terminus between ARF5 and ARF6. As shown in Fig. 3a, the fusion protein of the N-terminus of ARF6 and the C-terminus of ARF5 (N6C5), as well as ARF6, restored the PM localization of EGFR in *ARF6*$^{-/-}$ cells, but the N5C6 mutant or ARF5 failed to do so, indicating that the N-terminus is what makes ARF6 special.

We then mutated each of the four N-terminal amino acids (GKVL) in ARF6 to alanine and studied the function of the mutants. While the V4A and L5A mutants functioned as well as WT ARF6 to restore the PM localization of EGFR in *ARF6*$^{-/-}$ cells, G2A and K3A failed to do so (Fig. 3b and Supplementary Fig. 4c). But unlike G2A, K3A showed clear colocalization with EGFR in Golgi (Fig. 3b), suggesting that K3A might be able to bind EGFR but unable to lead to EGFR budding from Golgi.

To further confirm the results, we fused the WT or mutant N-terminus of ARF6 with GFP and examined whether they could compete with ARF6 in WT cells. While the WT version of the fusion protein GKVL-GFP, the K3A mutant GAVL-GFP, the V4A mutant GKAL-GFP, and L5A mutant GKVA-GFP effectively disrupted the PM localization of EGFR, the G2A mutant AKVL-GFP failed to do so (Fig. 3c and Supplementary Fig. 4d), implying that Lys3, Val4, and Leu5, unlike Gly2, might not be required for the binding of ARF6 to EGFR. Indeed, when subjected to immunoprecipitation, unlike the G2A mutant, the K3A mutant showed comparable interaction with EGFR as WT ARF6 (Fig. 3d).

### Lys3 of ARF6 recruits the exocyst complex to facilitate EGFR budding from Golgi

The results above suggest that Lys3 of ARF6 might be essential for EGFR budding from Golgi, possibly by recruiting a key factor in cargo budding. To search for the protein that binds Lys3 of ARF6, we co-expressed EGFR-Flag and ARF6-HA in HEK293T cells and performed tandem purification. Nocodazole was used to disrupt microtubules and vesicle trafficking to enrich the EGFR/Arf6 complex and its interacting partners at the Golgi complex. After mass spectrometry analysis, the exocyst complex subunits attracted our attention (Fig. 4a and Supplementary Data 1), as it is essential for vesicle trafficking to PM[22,23]. Indeed, the knockdown of three of the subunits, EXOC2, EXOC5, or EXOC6, all disrupted the PM localization of EGFR (Fig. 4b).

To map the subunit of the exocyst complex that binds ARF6, we co-expressed each of the eight subunits with ARF6 in HEK293T cells and found that EXOC2 showed the strongest interaction with ARF6 (Fig. 4c). And we showed that N6C5, but not N5C6, showed strong interaction with EXOC2 (Fig. 4d), indicating that the N-terminus of ARF6 was recognized by EXOC2. Furthermore, G2A and K3A showed dramatically decreased binding affinity with EXOC2, as demonstrated

by both immunoprecipitation and in vitro binding analysis (Fig. 4e, f). Considering that Gly2 myristoylation is essential for exposing the N-terminus of ARFs[24] and that Gly2 itself is engaged in binding palmitoylated EGFR, it is very likely that Lys3 of ARF6 is the amino acid that directly binds EXOC2.

### EFA6B converts ARF6 to GTP-bound form to facilitate EGFR transporting to PM

As small GTPases exist in either a GDP- or GTP-bound form[25], we asked which form of ARF6 was required for its activity. We used the T27N and Q67L mutants to mimic the GDP- and GTP-bound forms, respectively. When introduced into *ARF6*$^{-/-}$ cells, Q67L, but not T27N, restored the PM localization of EGFR (Fig. 5a). Furthermore, a fast cycling mutant of ARF6, T157A[26], also restored the PM localization of EGFR (Supplementary Fig. 5), indicating that the GTP-bound form ARF6 is required for transporting EGFR to PM. Notably, in T27N expressing cells, EGFR budded out from Golgi, colocalized with T27N in the puncta-like structures in the cytosol, but failed to reach PM (Fig. 5a), indicating that the GTP-bound form of ARF6 is not required for EGFR budding from Golgi, but indispensable for the subsequent transporting to PM. Compared to WT and Q67L, T27N showed decreased interactions with EXOC2 (Fig. 5b), suggesting that activation of ARF6 enhances its association with EXOC2 and accelerates the transport of EGFR.

We then went on to identify the guanine nucleotide exchange factor (GEF) of ARF6. Currently, there are eight reported GEFs of ARF6, Cytohesin1-3, EFA6A-D, and GEP100[27]. As EFA6A and EFA6C are exclusively expressed in the brain[27], we knocked down each of the other six GEFs in HeLa cells. Figure 5c showed that the knockdown of EFA6B, but not the other five GEFs, disrupted the PM localization of EGFR. Furthermore, the RUSH analysis revealed that EFA6B was recruited after the budding of EGFR from Golgi to facilitate vesicle transporting to PM (Fig. 5d).

To summarize our findings above, we proposed a working model of how the ARF6-mediated sorting system targets palmitoylated EGFR from Golgi to PM (Fig. 5e). When the newly synthesized EGFR gets palmitoylated at Golgi, and the palmitoylation is recognized by the N-myristoylation of ARF6 via lipid–lipid interaction. EXOC2 then binds Lys3 of ARF6 and recruits the exocyst complex to facilitate EGFR budding from Golgi. After that, EFA6B is recruited to EGFR-containing vesicles to convert ARF6 to GTP-bound form to facilitate vesicle transporting to PM.

### Knockdown of ARF6 inhibits the growth of EGFR-overexpression tumors

We then examined whether we could target the sorting to treat tumors with aberrant activation of EGFR signaling. We first examined the requirement of ARF6 on the PM localization of endogenous WT EGFR in A549 and MDA-MB-231 cells, both of which have EGFR overexpression. As shown in Fig. 6a, knockdown of ARF6 disrupted the PM

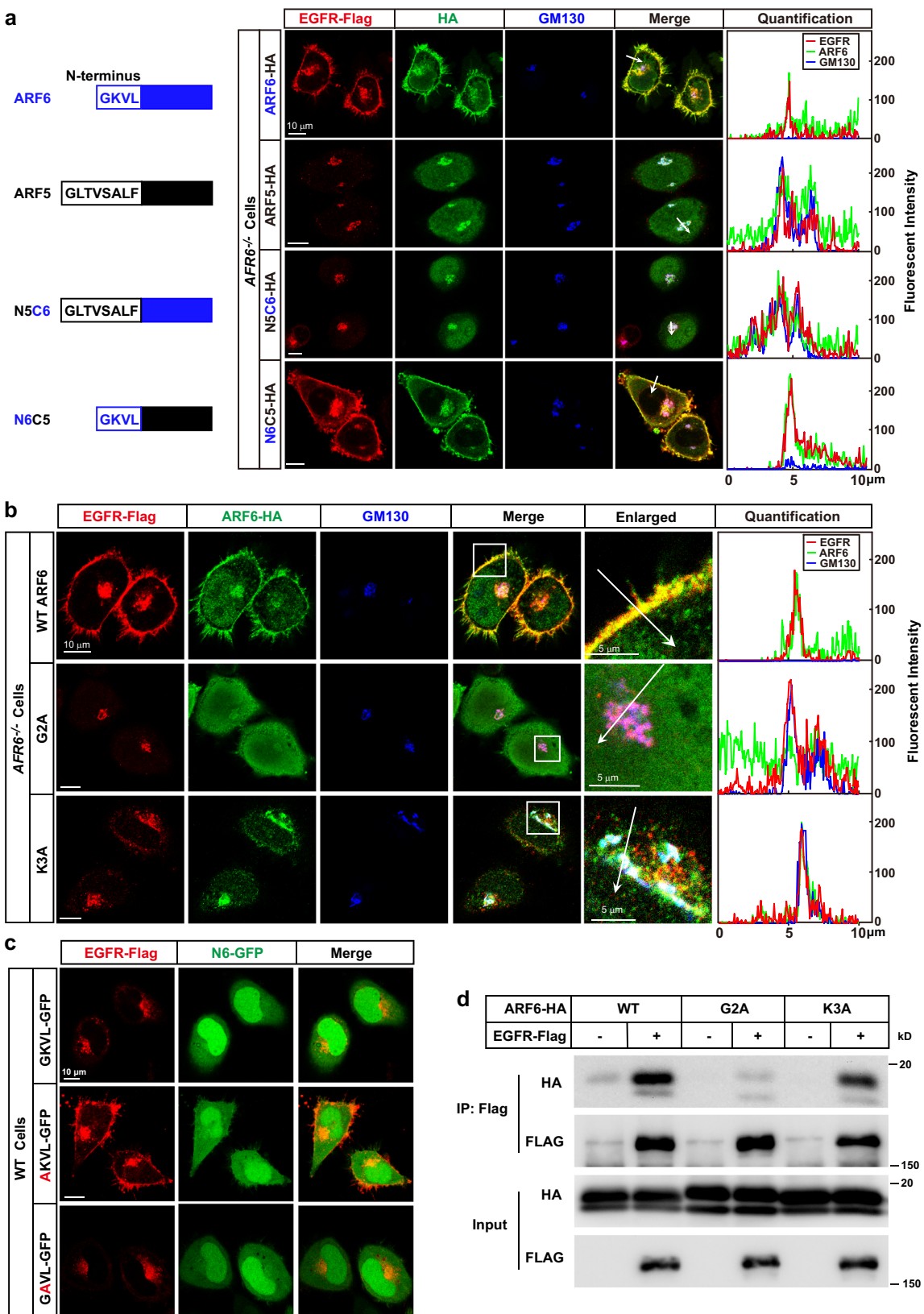

localization of EGFR in both cell lines. ARF6 was also required for the PM localization of five common clinical EGFR mutants, including deletion mutants of aa746–750 (Del 746–750), L858R, Del 746–750/T790M, L858R/T790M, and T790M/C797S (Supplementary Fig. 6a). And knockdown of ARF6 blocked EGF-induced EGFR signaling in A549 and MDA-MB-231 cells (Fig. 6b). Furthermore, EGF induced pEGFR in PC9 (Del 746–750) and H1975 (L858R/T790M) were also dramatically decreased in ARF6 knockdown cells (Supplementary Fig. 6b). Notably, knockdown of ARF6 dramatically decreased the total protein levels of EGFR (Fig. 6b and Supplementary Fig. 6b), suggesting that failure of

**Fig. 3 | The N-terminus of ARF6 is required for EGFR budding from Golgi. a** On day 0, ARF6$^{-/-}$ HeLa cells were set up and transfected with EGFR-Flag/ pCDH-puro and HA-tagged ARF6, ARF5, N5C6, or N6C5 on pCDH-puro (0.5 μg each). On day 2, cells were serum-starved overnight. On day 3, cells were fixed and subjected to immunostaining using anti-HA, anti-Flag, and anti-GM130 antibodies. Quantification of the fluorescent intensity was performed using ZEN2.3. The domain structures of the ARF proteins were illustrated in the left panel. Arrows indicate the area that was used for quantification. **b** ARF6$^{-/-}$ HeLa cells were set up, transfected, and subjected to immunostaining as in (**a**), except that different mutants of ARF6-HA/ pCDH-puro were used. On day 3, cells were harvested and subjected to immunostaining using anti-HA, anti-Flag, and anti-GM130 antibodies. **c** WT HeLa cells

were set up and transfected with EGFR-Flag/pCDH-puro and WT or mutants of the N-terminus of ARF6 fused with GFP (N6-GFP)/pCDH-puro. On day 3, cells were harvested and subjected to immunostaining using an anti-Flag antibody. Scale bar, 10 μm. **d** HEK293T cells were set up and transfected with indicated plasmids. On day 2 (8 h after transfection), cells were treated with 1 μM nocodazole overnight. On day 3, after the removal of nocodazole for 0.5 h, cells were treated with 2 mM DSP in ice-cold PBS for 2 h. The crosslink was stopped by Tris (pH 8.0, 0.1 M) at 4 °C for 15 min. Cells were lysed and subjected to immunoprecipitation using an anti-Flag antibody. Input and pellet fractions were subjected to western blot using indicated antibodies. Scale bars were as indicated. Source data are provided as a Source Data file.

sorting EGFR to PM caused its degradation. We then examined the half-life of EGFR in control and ARF6 knockdown A549 cells. Figure 6c, d showed that the knockdown of ARF6 significantly accelerated the degradation of EGFR and increased the ubiquitination of endogenous EGFR. These results indicate that disrupting the sorting of EGFR to PM results in its ubiquitination and subsequent degradation.

We then explored the role of ARF6 in tumor growth and found that knocking down ARF6 significantly inhibited the proliferation of WT EGFR cells, A549 cells (Fig. 6e), and MDA-MB-231 (Supplementary Fig. 6c), as well as EGFR mutant cells, PC9 and H1975 cells (Supplementary Fig. 6d, e). To further verify that ARF6 controls cell proliferation via EGFR, we also knocked down EGFR in A549 and MDA-MB-231 cells. Similar to the knockdown of ARF6, the knockdown of EGFR also significantly decreased cell proliferation, and double knockdown did not show further effect (Fig. 6f and Supplementary Fig. 6f).

We next tested the effect of ARF6 on the growth of xenograft tumors. As EGFR mutant tumors can be targeted by EGFR-specific TKIs or monoclonal antibodies[1], tumors overexpressing WT EGFR still lack effective strategies; we, therefore, focused on EGFR-overexpression tumors. We implanted control and ARF6 knockdown A549 or MDA-MB-231 cells into nude mice, and found that the knockdown of ARF6 significantly slowed the growth of these tumors (Fig. 6g and Supplementary Fig. 6g). At the end of the experiments, the mass of ARF6 knockdown tumors were also significantly lower than control tumors (Fig. 6h, i and Supplementary Fig. 6h, i).

### Myristoylated GKVL-TAT inhibits the growth of EGFR-overexpression tumors

We then sought to find an inhibitor of the ARF6-mediated sorting system to target EGFR-overexpression tumors. As we showed that fusing the N-terminus of ARF6 with GFP disrupted the sorting system (Fig. 3c), we synthesized a myristoylated N-terminal peptide of ARF6 (GKVL) with an HIV-TAT sequence at the C-terminus to make it cell-permeable and designated the peptide as Myr-GKVL-TAT (Fig. 7a, upper panel). We have also synthesized a peptide with the same sequence but without the N-terminal myristoylation as a control, designated as GKVL-TAT. We first tested the effect of these two peptides in A549 and MDA-MB-231 cells and found that treatment with Myr-GKVL-TAT, but not the non-myristoylated GKVL-TAT, disrupted PM localization of EGFR in both cell lines (Fig. 7a and Supplementary Fig. 7a). And the Myr-GKVL-TAT peptide dramatically decreased EGF-induced EGFR signaling (Fig. 7b and Supplementary Fig. 7b). The Myr-GKVL-TAT peptide effectively inhibited cell viability of A549 cells and MDA-MB-231 cells at an IC$_{50}$ of 22.7 and 31.6 μM, respectively (Fig. 7c and Supplementary Fig. 7c).

To further evaluate the potential therapeutic value of the Myr-GKVL-TAT peptide, we implanted A549 or MDA-MB-231 cells into nude mice, let the tumor grow to about 100 mm³, and started to treat the tumors with a daily injection of vehicle, Myr-GKVL-TAT or GKVL-TAT. Compared with vehicle and GKVL-TAT, Myr-GKVL-TAT significantly inhibited the growth of both tumors (Fig. 7d and Supplementary

Fig. 7d). At the end of the experiment, the tumor mass in the Myr-GKVL-TAT-treated group was significantly reduced compared to the other two groups (Fig. 7e, f and Supplementary Fig. 7e). The EGFR signaling was dramatically decreased in Myr-GKVL-TAT-treated tumors (Supplementary Fig. 7f, g). These results indicated that targeting the sorting system by the Myr-GKVL-TAT peptide represents a promising strategy to treat EGFR-dependent tumors.

## Discussion

Our findings have provided a potent strategy to target EGFR-dependent tumors. Currently, there are mainly two strategies to treat EGFR-driven tumors, anti-EGFR monoclonal antibodies that block dimerization of EGFR and EGFR-targeted TKIs that directly inhibit the kinase activity[28,29]. These strategies are effective for treating tumors bearing EGFR mutations but not for tumors with overexpression of WT EGFR[30,31]. Here, we identify EGFR as a cargo of the ARF6-mediated sorting system for palmitoylated proteins. Knocking down ARF6 blocks EGFR sorting to PM, causes degradation of EGFR, disrupts EGF-induced EGFR signaling, and inhibits the growth of EGFR-overexpression tumor cells. Furthermore, based on the sorting mechanism, we designed a cell-permeable peptide, Myr-GKVL-TAT, and found it dramatically inhibited the growth of two EGFR-overexpression tumors. It is worth mentioning that knocking down ARF6 also disrupted the PM localization of EGFR mutants and inhibited the growth of tumor cells. And acquired resistance is ubiquitously developed and limits TKIs efficiency in EGFR mutant tumors with TKIs therapy[3–5]. Therefore, blocking the ARF6-mediated sorting system not only represents a potential strategy to target EGFR-overexpression tumors, but also might be an alternative strategy to treat EGFR mutant tumors.

Notably, EGFR has also been shown to regulate the activation of ARF6. In breast cancers, GEF100, another GEF protein of ARF6, has been shown to bind Tyr1068/1086-phosphorylated EGFR to activate Arf6 to induce TNBC invasion and metastasis[32]. EGFR also activates ARF6 to stimulate oncogenic Ras tumor overgrowth by regulating Hh signaling[33]. In fact, MDA-MB-231 and A549 cells used in our experiments also bear KRas mutations[16,17], and we show that knockdown of either EGFR or ARF6 inhibits tumor growth. Combining the previous observations with our results, we propose that ARF6 and EGFR might regulate the activity of each other to promote tumor progression by forming a positive feedback loop. On the one hand, ARF6 controls EGFR signaling by transporting EGFR to PM; on the other hand, EGFR on PM binds its ligands to activate ARF6.

Targeting other components in the sorting system, like inhibiting EGFR palmitoylation or EFA6B, might also be an effective strategy to treat EGFR-driven tumors. We show that inhibiting the palmitoylation of EGFR by 2-BP blocks the sorting of EGFR to PM. We further identify that DHHC13 palmitoylates EGFR at Cys775, Cys781, and Cys797 and is required for EGFR PM localization and signaling. In agreement with our studies, a previous study shows that treatment with 2-BP or fatty acid synthase inhibitor cerulenin disrupted the PM localization of EGFR, reduces EGFR protein levels and signaling, and sensitizes A549 and PC3

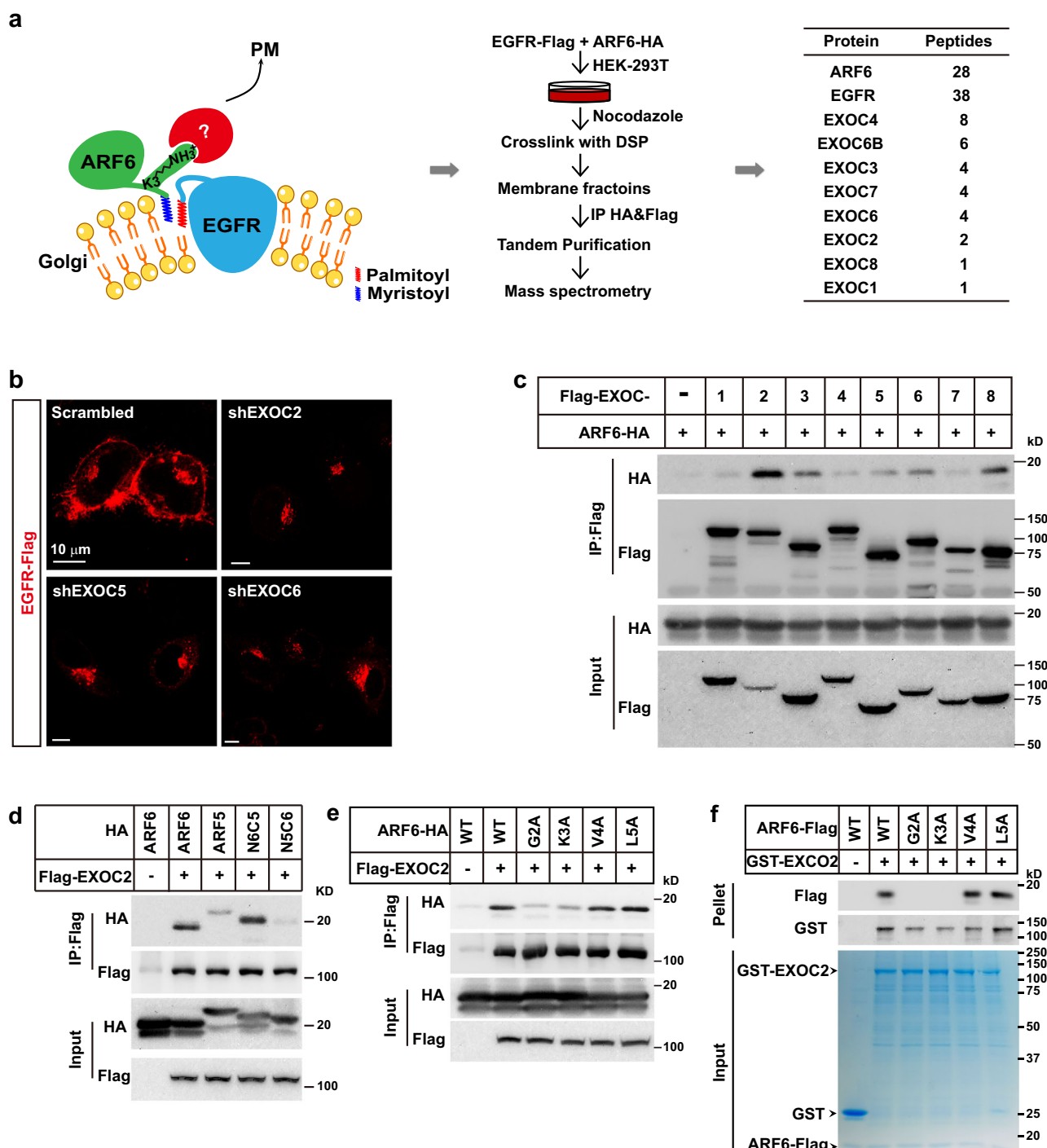

**Fig. 4 | Lys3 of ARF6 recruits the exocyst complex to facilitate EGFR budding from Golgi. a** On day 0, HEK293T cells were set up at $7 \times 10^5$ cells per 10-cm dish. On day 2, cells were transfected with EGFR-Flag/pCDH-puro and ARF6-HA/pCDH-puro (4 μg each). On day 3, cells were treated and subjected to tandem affinity purification using anti-HA and anti-Flag conjugated beads as described in Methods. The eluted fraction was analyzed by mass spectrometry. **b** On day 0, HeLa cells were set up as in Fig. 1c. On day 1, cells were infected with lentivirus expressing shRNAs of EXOC2, EXOC5, or EXOC6 on a pLL3.7 vector. On day 3, cells were transfected with EGFR-Flag/pCDH-puro. On day 4, cells were harvested for immunostaining. Scale bar, 10 μm. **c** HEK293T cells were set up and transfected with ARF6-HA/pCDH-puro and each of the Flag-tagged subunits of the Exocyst complex on a pcDNA3.3 vector. On day 3, cells were harvested for immunoprecipitation using anti-Flag M2 beads. The input and pellet fractions were subjected to western blot using indicated antibodies. **d** HEK293T cells were co-transfected with Flag-EXOC2 and HA-tagged

ARF6, ARF5, N6C5, or N5C6, harvested for immunoprecipitation with anti-Flag M2 beads and subjected to western blot. The input and pellet fractions were subjected to western blot using indicated antibodies. **e** HEK293T cells were set up, co-transfected with Flag-EXOC2/pCDH-puro and N-terminal mutants of ARF6, and subjected to immunoprecipitation as in (**c**). The input and pellet fractions were subjected to western blot using indicated antibodies. **f** WT and mutants of ARF6-Flag were expressed and purified from HEK293T cells. GST and GST-EXOC2 were bacterially expressed and purified. The proteins with indicated combinations were incubated and subjected to GST-pull down analysis as described in Methods. The input was subjected to Coomassie staining, and the pellet fractions were subjected to western blot using the indicated antibodies. Source data are provided as a Source Data file.

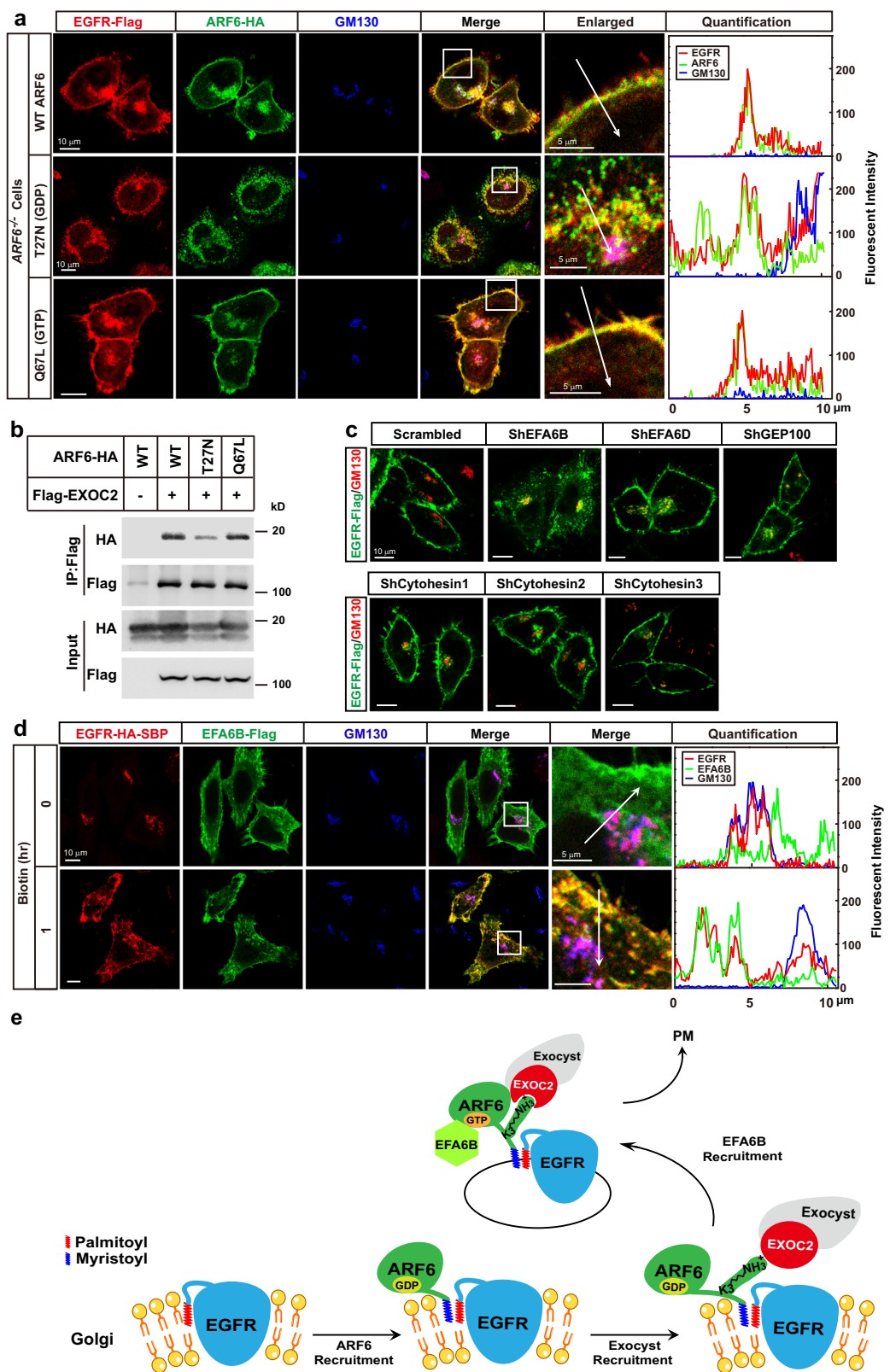

cells to TKIs, and that palmitoylation of Cys797 is the key player in the process[34]. In another study, Cys1025 and Cys1122 of EGFR are found to be palmitoylated and play different roles in its function, but inhibiting palmitoylation of EGFR by 2-BP also sensitizes MDA-MB-231 cells to TKIs[14]. Furthermore, treatment with 2-BP inhibits the growth of NSCLC-bearing EGFR or KRAS mutations[15,35]. Therefore, although 2-BP is a non-specific inhibitor of palmitoylation and is unlikely to be applied to

clinical studies, targeting EGFR palmitoylation represents a potential strategy to treat EGFR-dependent tumors. Similarly, inhibitors of EFA6 or small molecules that specifically block the binding of ARF6 to EFA6B might also be an attractive strategy.

In addition, our findings have also provided valuable insights into how palmitoylation directs protein sorting. Although accumulating evidences have demonstrated that palmitoylation functions as a

**Fig. 5 | EFA6B converts ARF6 to GTP-bound form to facilitate EGFR transporting to PM. a** *ARF6*−/− HeLa cells were set up, transfected, and subjected to immunostaining and quantification as in Fig. 3a, except that different ARF6 mutants were used. On day 3, cells were harvested for immunostaining with anti-Flag, anti-HA, and anti-GM130 antibodies. Arrows indicate areas used for the quantification of fluorescent intensity. Scale bar, 10 μm. **b** On day 0, HEK293T cells were transfected with Flag-EXOC2 and WT, T27N or Q67L mutant of ARF6-HA. On day 1, cells were lysed and subjected to immunoprecipitation with anti-Flag M2 beads. The input and pellet fractions were subjected to western blot using indicated antibodies. **c** On day 0, HeLa cells infected with indicated shRNAs were set up as in Fig. 4b. On day 2, cells were transfected with EGFR-Flag/pCDH-puro. On day 3, cells were harvested and subjected to immunostaining using anti-Flag and anti-GM130 antibodies. Scale bar, 10 μm. **d** WT HeLa cells were set up, transfected with indicated plasmids, and subjected to RUSH analysis as in Fig. 2b. Arrows indicate areas used for quantification of fluorescent intensity. **e** A schematic model to summarize how the ARF6-mediated sorting system directs the sorting of palmitoylated EGFR from Golgi to PM. Source data are provided as a Source Data file.

sorting signal to direct proteins to destined membranes[11,12], little is known about the sorting machinery and the underlying mechanisms. Here, we report sorting machinery that transports palmitoylated EGFR from Golgi to PM. In the sorting system, ARF6 plays a central role. First, ARF6 recognizes palmitoylated EGFR through lipid–lipid interaction. Second, ARF6 is required for the EGFR budding from Golgi. We show that Lys3 of ARF6 is recognized by EXOC2 and thus recruits the exocyst complex to promote EGFR budding from Golgi. Third, ARF6, in the GTP-bound form, is required for transporting the EGFR-containing vesicles to PM. We show that EFA6B is recruited to the vesicles to convert ARF6 to GTP-bound form to facilitate vesicle trafficking to PM. Notably, the GDP-bound form is enough for ARF6 to fulfill its functions in the two steps above, as neither the GDP-form mimicking T27N mutant of ARF6 nor the knockdown of ARF6 GEF protein EFA6B affects EGFR budding from Golgi.

## Methods

Our research complies with all relevant ethical regulations of Xiamen University and Fudan University.

### Stock preparation

Stock solutions of nocodazole (1 mM, MCE, HY-13520), D-Biotin (200 mM, Sangon Biotech, A600078), and DSP (Thermo Scientific, 22586) were made up in DMSO. A stock solution of hydroxylamine HCl (2 M, Sigma-Aldrich, 159417) was freshly prepared in water and was adjusted to pH 7.5 with NaOH.

### Plasmids

Full-length cDNA of human ARF1, ARF3, ARF4, ARF5, *ARF6*, *EXOC1-8*, *EFA6B*, and *EGFR* were cloned from a cDNA library prepared from HEK293T or HeLa cells. These genes were cloned into either pcDNA3.3 or pCDH-EF1-MCS-IRES-Puro (pCDH-puro, System Biosciences) with indicated tags. For knockdown, shRNAs were cloned into pLKO.1 (Addgene, 10878) or pLL3.7 (Addgene, 11795). SgRNA was cloned into pLentiCRISPR (Addgene, 52961). Site-directed mutagenesis was performed using commercial kits from New England Biolabs. The primer sequences are listed in Supplementary Table 1.

### Cell culture

HEK293T, HeLa, MDA-MB-231, and A549 cells were cultured in Medium (Dulbecco's modified Eagle's medium (4.5 g/L glucose) supplemented with 10% (v/v) FCS (Thermo Fisher Scientific), 100 U/ml penicillin, and 100 mg/ml streptomycin) at 37 °C in an atmosphere of 5% CO_2. Cell viability was determined by cell counting kit-8 (Topscience, C0005).

### Lentivirus production and infection

For lentivirus packaging, indicated gene on pCDH-puro, shRNA on pLKO.1, or sgRNA on pLentiCRISPR was co-transfected with psPAX2 and pMD2.G into HEK293T cells as described[36]. For infection, cells were infected at 50–70% confluence with lentivirus in a medium containing 8–10 μg/ml polybrene. Cells were selected against 1 μg/ml (HeLa, MDA-MB-231, A549) puromycin for at least 48 h before use for the described experiments.

### Generation of *ARF6*−/− HeLa cells

*ARF6*−/− HeLa cells were generated with the CRISPR/Cas9 system. Two sgRNAs flanking the coding regions of *ARF6* were designed (Supplementary Table 1) and cloned into pLentiCRISPR for lentivirus packaging. HeLa cells were infected with lentivirus encoding these two sgRNAs and selected with 1 μg/ml puromycin. Cells were then seeded into 96-well plates and single clones were tested to confirm the knockout of ARF6. A subclone of *ARF6*−/− cells was used for the study.

### Immunofluorescence

Immunofluorescence was performed as previously described[36]. Images were taken using ZEN2010 on Zeiss LSM-780 confocal microscopy. Co-localization analysis was performed using ZEN2.3 software. The following primary antibodies were used: anti-Flag (1:200, Sigma-Aldrich, F3165), anti-HA (1:200, Roche, 11867423001), anti-GM130 (1:200, Cell Signaling Technology, #12480 s), anti-GM130 (1:200, BD, 610822), and anti-EGFR (1:100, Proteintech, 18986-1-AP). The secondary antibodies were: Donkey anti-Mouse IgG Alexa Fluor 488 (Thermo Fisher Scientific, A21202), Goat anti-mouse IgG Alexa Fluor 647 (Thermo Fisher Scientific, A21236), Goat anti-Rat IgG Alexa Fluor 594 (Thermo Fisher Scientific, A11007), Donkey anti-Rabbit IgG Alexa Fluor 647 (Thermo Fisher Scientific, A31573), and Goat anti-Rabbit IgG Alexa Fluor Plus 555 (Thermo Fisher Scientific, A21429). FITC or Rhodamine-labeled phalloidin (1:80, ABclonal Technology, RM02836, RM02835) were used to stain F-actin to indicate localization of plasma membrane. For the Proximity ligation assay, Rabbit anti-HA antibody (1:200, Cell Signaling Technology, #3724) and mouse anti-Flag antibody (1:200, Sigma-Aldrich, F3165) were used as primary antibodies. Duolink® In Situ Red Starter Kit Mouse/Rabbit (DUO92101, Sigma-Aldrich) was used following the manufacturer's instructions.

### Western blot

The following primary antibodies were used: anti-Flag (Sigma-Aldrich, F3165), anti-HA (1:1000, Roche, 11867423001), anti-ARF6 (1:1000, Proteintech, 20225-1-AP), anti-EGFR (1:1000, Proteintech, 18986-1-AP), anti-phospho-EGFR (Tyr1068) (1:1000, Cell Signaling Technology, 3777 s), anti-Akt (1:1000, Proteintech, 10176-2-AP), anti-phospho-Akt (Ser473) (1:1000, Cell Signaling Technology, 4060 s), anti-ERK (1:1000, Cell Signaling Technology, 4695 s), anti-phospho-ERK (Thr202/Tyr204) (1:1000, Cell Signaling Technology, 4370 s), anti-GAPDH (1:10000, Proteintech, 60004-1-Ig), anti-Ub (P4D1) (1:1000, Santa Cruz, sc-8017). Blots were developed in the ChemStudio imaging system (Analytik Jena AG).

The VisionWorks software on a ChemStudio imaging system was used for the collection and quantification of western blot images.

### Detection and isolation of palmitoylated proteins

Palmitoylated proteins were analyzed by acyl-resin assisted capture assay (Acyl-RAC) as previously described[36]. Input and pellet fractions were analyzed on SDS-PAGE and blotted with indicated antibodies.

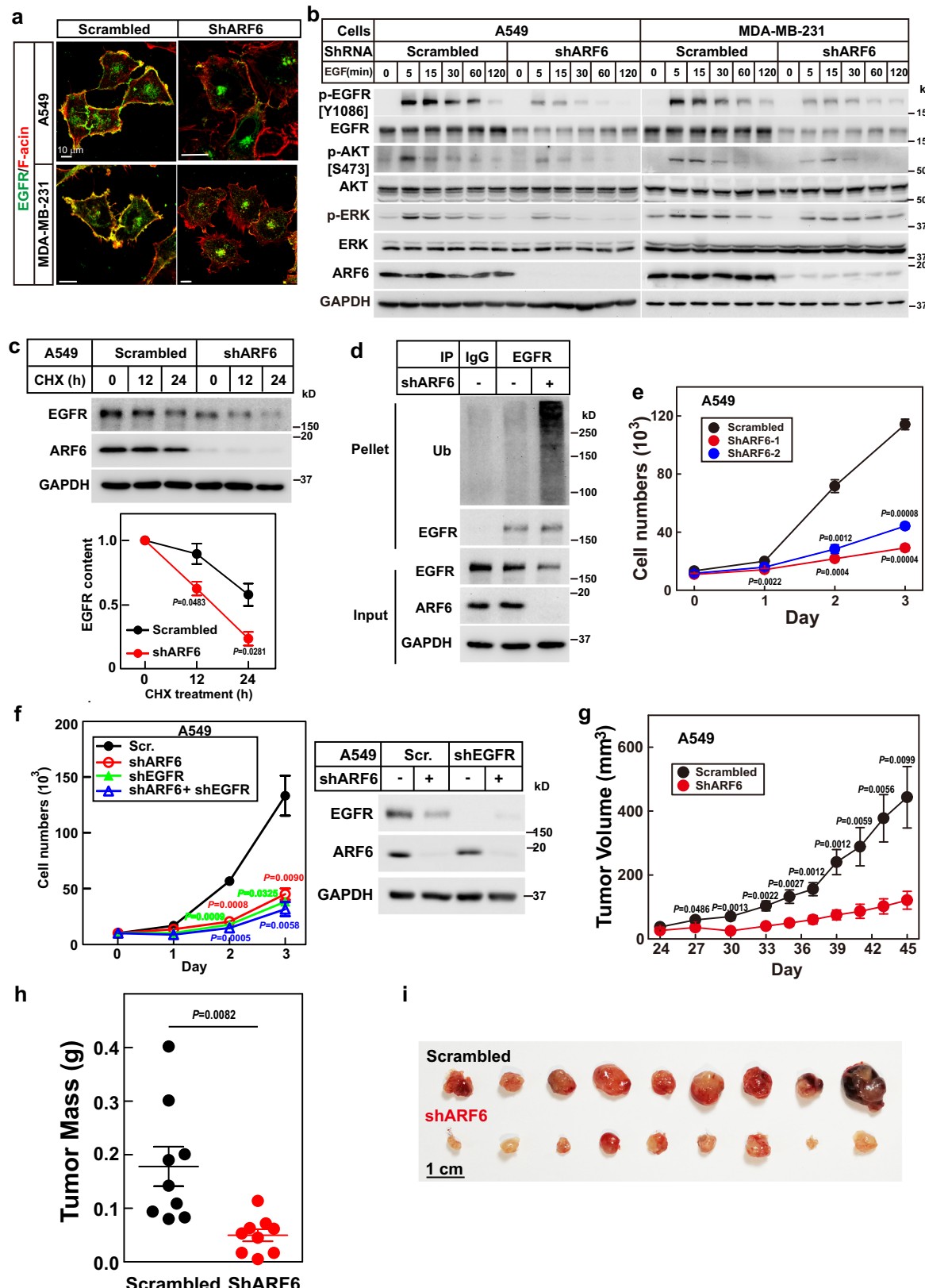

## Identification of the palmitoylation sites of EGFR by mass spectrometry

On day 0, HEK293T cells were transfected with EGFR-Flag. On day 2, cells were harvested with Buffer A (100 mM Hepes, pH 7.4, 0.1% SDS) containing 150 mM NaCl and protease inhibitors, and subjected to immunoprecipitation with anti-Flag M2 beads (Sigma) for 4 h. The beads were washed with Buffer A containing 500 mM NaCl three times and eluted with 0.4 mg/ml 3 × Flag peptide (in Buffer A, 150 mM NaCl). 10% SDS (2.5%, final concentration) and TCEP (10 mM, final concentration) were added to the eluted fraction and rotated at room temperature for 10 min. N-ethyl maleimide (NEM, Sigma) was added to 25 mM to block free Cys at room temperature

**Fig. 6 | Knocking down ARF6 inhibits the growth of EGFR-overexpression tumors. a** Control and ARF6 knockdown A549 or MDA-MB-231 cells were set up as in Fig. 1a. Cells were serum-starved overnight and subjected to immunostaining using an anti-EGFR antibody. Scale bar, 10 μm. **b** Control and ARF6 knockdown A549 or MDA-MB-231 cells were set up at $1.2 \times 10^5$ cells per 35-mm dish, serum-starved overnight, and treated with EGF (100 ng/ml) for the indicated time. **c** Control and ARF6 knockdown A549 cells were serum-starved, treated with cycloheximide (100 μg/ml) for the indicated time, and harvested for immuno-blotting. Band intensity of EGFR was quantified using Visionworks on the Chem-studio imaging system and showed as mean ± SEM of a triplicate. The statistical analysis was performed by a two-tailed paired *t*-test. **d** Control and ARF6 knock-down A549 cells were serum-starved, pretreated with chloroquine (50 μM) for 6 h, and harvested for immunoprecipitation using an anti-EGFR antibody. **e** On day 0, control and ARF6 knockdown A549 cells were set up at $1 \times 10^4$ cells per 35-mm dish.

From day 1 to day 3, cells were harvested and counted each day. Each value represents the mean ± SEM of a triplicate. *P* value denotes the level of statistical significance (two-tailed paired *t*-test). **f** A549 cells expressing indicated shRNAs were set up as in (**e**) and harvested for cell count each day. Each value represents the mean ± SEM of a triplicate. *P* value denotes the level of statistical significance (two-tailed paired *t*-test) between scrambled and indicated shRNA-expressing cells. Protein levels of EGFR and ARF6 in A549 cells were confirmed by western blot in the right panel. **g–i** Control and ARF6 knockdown A549 cells were subcutaneously implanted into 6-week-male nude mice at $3 \times 10^6$ cells per spot. Starting from day 24, tumor size was measured every three days (**g**). On day 45, mice were dissected, tumor masses were weighted (**h**), and images were taken (**i**). Each value represents the mean ± SEM of nine samples. The statistical analysis was performed by a student's two-tailed paired *t*-test. Source data are provided as a Source Data file.

for 2 h. Proteins were precipitated with cold acetone at −20 °C overnight, followed by three washes of 70% cold acetone. The pellet was then resuspended in Buffer B (100 mM Hepes, pH 7.5, 1 mM EDTA, and 1% SDS), incubated with thiopropyl Sepharose 6B beads with 0.5 M NH₂OH and rotated at 37 °C for 3 h. After five washes with Buffer B containing 8 M urea and two washes with Buffer B without urea, proteins were eluted with Buffer B containing 100 mM DTT at 37 °C for 30 min, boiled in sample loading buffer, and separated by SDS-PAGE.

After staining of gels with Coomassie blue, the band containing EGFR was excised, and it was reduced with 10 mM TCEP and alkylated with 40 mM iodoacetamide at 60 °C for 30 min, followed by trypsin digestion. Samples were analyzed on a nanoElute (Bruker) coupled to a timsTOF Pro (Bruker) equipped with a CaptiveSpray source. Peptides were dissolved in 10 μL 0.1% formic acid and were auto-sampled directly a homemade C18 column (35 cm × 75 μm i.d., 1.9 μm 100 Å). Samples were eluted for 60 min with linear gradients of 3–35% acet-onitrile in 0.1% formic acid at a flow rate of 300 nL/min. Mass spectra data were acquired with a timsTOF Pro mass spectrometer (Bruker) operated in PASEF mode. The raw files were analyzed by Peaks Studio X software against the Uniprot database.

### Retention using a selective hook (RUSH)
RUSH system[19] was used to study the sorting of EGFR from Golgi to PM. The coding region of a Golgi hook, the fusion protein of streptavidin and Golgin-84 (Addgene, 65305), was cloned into pCDH-puro. On day 0, Streptavin-Golgin-84/pCDH-puro, WT or 9CS mutant of EGFR-HA-SBP/pCDH-puro, WT or mutant ARF6-Flag/pCDH-puro and other indicated plasmids were co-transfected with into WT or *ARF6*⁻/⁻ HeLa cells. On day 1, cells were treated with 100 μM biotin and harvested at various times for immuno-fluorescence analysis.

### Identification of EGFR/ARF6 associated proteins
On day 0, HEK293T cells were transfected with EGFR-Flag and ARF6-HA. Eight hours after transfection, cells were treated with 1 μM noco-dazole overnight. On day 1, cells were washed twice with PBS and incubated in a fresh medium for another 0.5 h. Cells were then washed and harvested in PBS and treated with 2 mM DSP in ice-cold PBS for 2 h. The crosslink was stopped by Tris (pH 8.0, 0.1 M) at 4 °C for 15 min. Cells were resuspended in Buffer C (10 mM Tris pH 7.5, 10 mM KCl, complete protease inhibitor cocktail (MCE)) and dounce homo-genized. After centrifugation at $1000 \times g$ for 10 min at 4 °C, the resul-tant supernatant was spun at $100,000 \times g$ for 1 h at 4 °C to collect membrane fractions, which were dissolved in Buffer D (50 mM Tris, pH 7.5, 150 mM NaCl, 2% NP-40, 1% SDS) for 1 h. After centrifugation at $12,000 \times g$ for 10 min, the supernatant was diluted ten times with Buffer E (50 mM Tris pH 7.5, 150 mM NaCl, with protease inhibitor) and

subjected to immunoprecipitation with HA-antibody conjugated beads (MCE, HY-K0201) overnight. The beads were washed five times with Buffer E and eluted with 1 mg/ml HA peptide. The eluted proteins were then subjected to immunoprecipitation using anti-Flag M2 beads (Sigma, A2220) for 4 h, followed by washing with Buffer E and eluting with 0.2 mg/ml 3 × Flag peptide. The eluted proteins were treated with 100 mM DTT in SDS sampling buffer at 37 °C for 1 h before being subjected to SDS-PAGE and mass spectrometry analysis.

### GST-pull down analysis
Human EXOC2 was cloned into pGEX4T-1 and transformed into BL21 (DE3) for protein expression. Bacteria were induced by 0.4 mM IPTG at 25 °C for 8 h. GST and GST-EXOC2 were purified using Glutathione Sepharose 4B (Amersham Pharmacia) following the manufacturer's instructions. The purified proteins were dialyzed in Buffer F to remove GSH.

WT and various mutants of ARF6-Flag were transfected into HEK293T cells. Cells were harvested, lysed in Buffer F (50 mM Tris, pH 7.5, 150 mM NaCl, 0.1% SDS, 1% Triton X-100 with protease inhibitors), and incubated with anti-Flag M2 beads for 4 h at 4 °C. The beads were washed with Buffer F twice, three times with Buffer F containing 500 mM NaCl and then once with Buffer G (50 mM Tris, pH 7.5, 150 mM NaCl, 0.1% Triton X-100), followed by elution with 0.2 mg/ml 3 x Flag peptides.

On the day of the experiment, purified GST or GST-EXOC2 (500 ng) was mixed with 200 ng purified WT or mutant of ARF6-Flag in Buffer G, and incubated with BSA-blocked Glutathione Sepharose 4B for 1 h at 4 °C. Beads were washed with Buffer G 5 times, boiled with SDS sample buffer, and subjected to western blot.

### Xenograft model and subject details
BALB/c-Nude male mice were purchased from GemPharmatech (Nanjing, China). All mice were housed in colony cages at 25 °C with 12-h light/12-h dark cycles. The dark cycle began at 7 p.m. All animal studies were performed with the approval of the Institutional Animal Care and Research Advisory Committee at Xiamen University. In all experiments, maximal tumor sizes were not exceeded 1000 mm³.

For tumor growth assay, indicated cells in the logarithm phase were trypsinized and subcutaneously injected into 6-week-old nude mice at indicated cell numbers in the figure legends. Tumor sizes were measured using a digital caliper every other day. Tumor volume was calculated as $0.5 \times length \times width^2$.

To test the effect of Myr-GKVL-TAT and GKVL-TAT on tumor growth, $3 \times 10^6$ MDA-MB-231 cells or $1 \times 10^7$ A549 cells were sub-cutaneously injected to 6-week-old nude mice. When tumor sizes reach about 100 mm³, vehicle, Myr-GKVL-TAT, or GKVL-TAT peptide was subcutaneously injected at 2 mg/kg once a day. Tumor sizes were measured every other day, as above.

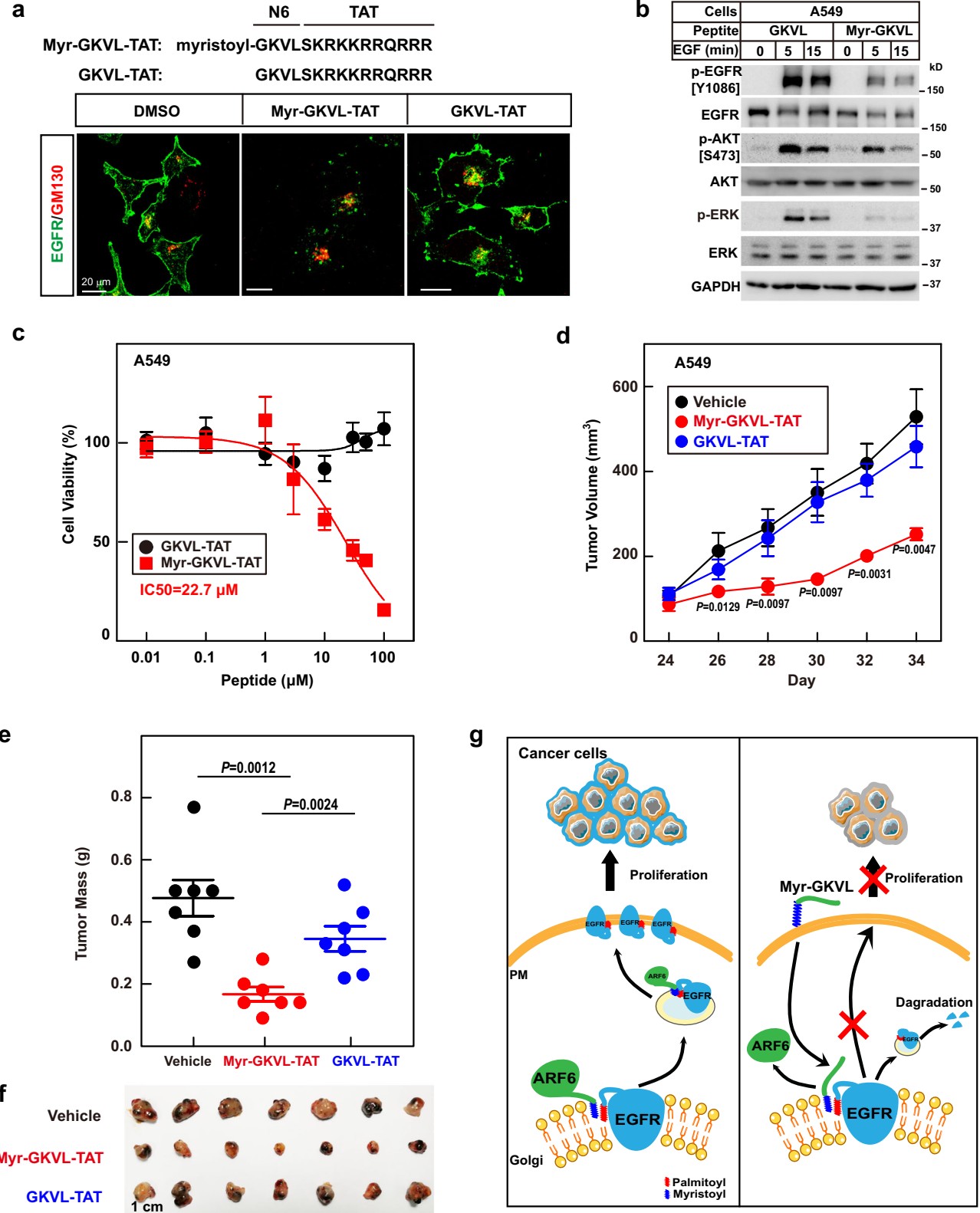

## Statistics and reproducibility

All the statistical analysis was performed by student's two-tailed paired *t*-test using EXCEL2010. All the values represent mean ± SEM. All the statistical details of the experiments can be found in the figure legends, including the exact number of cells or mice. No data were excluded from any of the experiments.

All the experiments were repeated at least twice, and the repeats are biological replicates.

## Reporting summary

Further information on research design is available in the Nature Research Reporting Summary linked to this article.

**Fig. 7 | The Myristoylated GKVL-TAT peptide inhibits the growth of EGFR-overexpression tumors. a**, **b** On day 0, A549 cells were set up as in Fig. 1a. On day 2, cells were treated with 10 μM Myr-GKVL-TAT or GKVL-TAT for 20 h. Cells were then incubated with serum-free medium, including the peptides, for 4 h.
**a** Immunostaining assays were performed using the anti-EGFR antibodies. The upper panel shows the sequence of the two peptides. Scale bar, 20 μm. **b** Cells were treated with 100 ng/ml EGF for the indicated time, harvested, and subjected to western blot. **c** On day 0, A549 cells were set up at $7 \times 10^3$ cells per well in a 96-well plate. From day 1, cells were treated with various concentrations of Myr-GKVL-TAT or GKVL-TAT for two days. On day 3, cell viability was determined by cell counting kit-8 (Topscience, C0005). Each value represents the mean ± SEM of a triplicate. The statistical analysis was performed by a student's two-tailed paired $t$-test. $IC_{50}$

was analyzed using Graphpad Prism 5. **d**–**f** On day 0, A549 cells were injected into nude mice subcutaneously at $1 \times 10^7$ cells per mouse ($n = 7$). Starting from day 24, when tumors grew to around 100 mm³, mice were treated with a daily subcutaneous injection of vehicle, Myr-GKVL-TAT (2 mg/kg), or GKVL-TAT (2 mg/kg). Tumor sizes were measured every 2 days (**d**). On day 34, mice were euthanized and tumors were dissected. Tumor weights were weighed (**e**), and representative images of mice and tumors were shown (**f**). Each value represents the mean ± SEM of seven samples. The statistical analysis was performed by a student's two-tailed paired $t$-test. **g** A working model shows ARF6 mediates palmitoylated EGFR sorting to PM, facilitating cancer cell proliferation. Source data are provided as a Source Data file.

## Data availability
All data generated or analyzed during this study are included in this published article and its supplementary information files. Source data are provided with this paper.

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

## Acknowledgements

This work was supported by the National Natural Science Foundation of China (32125022 and 92157301 to T.-J.Z., 32071150 to H.G., 31871193 to T.-J.Z, and 32100539 to J.W.), the National Key R&D Program of China (2020YFA0803601 to T.-J.Z.), and the China Postdoctoral Science Foundation (2021M690697 to J.W.).

## Author contributions

H.G., J.W., S.R., L.-F.Z., Y.-X.Z., D.-L.L., H.-H.S., L.-Y.L., C.X., and Y.-Y.W. performed the experiments. H.-R.W., X.D., and P.L. provided expertise and materials. H.G. and T.-J.Z. designed the experiments, analyzed the data, and wrote the manuscript.

## Competing interests

The authors declare no competing interests.
