## [Peer Review File · Nature Communications]

Targeting EGFR-dependent tumors by disrupting an ARF6-mediated sorting systemReviewers' comments:

Reviewer #1 (Remarks to the Author):

The findings reported by Guo et al are overall interesting. The concept of post-transductional modifications such as palmitoylation and myristoylation affecting trafficking and receptor function is interesting. However, from reading the paper, several issues can be raised to challenge the validity of the conclusions presented by the authors.

Fig 1: 2-BP treatment will affect the function of broad range of proteins. The use of such inhibitors does not prove the authors point. The function of the 9CS EGFR mutant has not been thoroughly characterized. Could misfolding of the receptor explain its retention in the Golgi?

Fig 2: ARF6 is the isoform found preponderantly at the plasma membrane. Why here, is it mainly intracellular? KD of ARF6 has been shown to regulate endocytosis of plasma membrane receptors. Here, it is unclear where ARF6 precisely acts. The use of overexpressed Flag-ARF6 (modified GTPase) might not represent the function of endogenously expressed ARF since modifications of the GTPase is known to alter function. The amount of overexpression might also impair localization and explain finding in 2C.

Fig 3: ARF5 and ARF6 chimeras are used, but no thorough characterization of the roles of all ARF isoforms either alone or in combination have been done.

Fig 5: The use of ARF mutants has to be done acknowledging the weaknesses these have. The consensus about the use of the constitutively active ARF6QL mutant is that inhibiting cycling might results in many cases as dominant negative effects. The only expression of this mutant alters cellular morphology. A fast cycling mutant needs to be tested.

Fig 6: The effect of ARF6 on EGFR function in TNBC cells is not novel. The data presented here do not give further insights.

Reviewer #2 (Remarks to the Author):

The manuscript by Huiling Guo et al. examines the requirement of the small GTPase Arf6 in trafficking EGFR from the Golgi apparatus to the plasma membrane. Disruption of this mechanism by shRNA targeting Arf6 blocks EGFR PM localization, EGFR signaling and tumor growth in xenograft assays. A myristoylated peptide from Arf6 also blocks tumor growth in mice. The authors propose palmitoylation of EGFR is required to be recognized by Arf6 which recruits the exocyst complex to mediate trafficking to the plasma membrane.

While it has been shown that the exocyst complex is involved in EGFR trafficking to the PM in stem cells and in kidney tissues, the involvement of Arf6 appears to be a new finding.

General Critique

My main concern with this manuscript is that the studies with EGFR all use an expression construct that has 9-cysteine residues mutated in the cytosolic domain. Has it been shown that EGFR still folds properly when 6 of the cysteine residues in the kinase domain are mutated or could it be unfolded/partially folded and trapped in the ER/Golgi compartments? If the protein is unfolded, it probably won't be palmitoylated

Most of the experiments in this study rely on 2-BP which is not a specific palmitoyl transferase inhibitor. It is a convenient first step towards showing a protein is palmitoylated, but the effect of 2BP on complex cellular processes cannot be relied on and needs verification by genetically inhibiting the relevant transferase. Runkle et al. *Molecular Cell* (2016) showed that EGFR is palmitoylated by DHHC20 and identified two palmitoylation sites in the unstructured C-terminal tail by mass spectrometry. Knocking down DHHC20 or mutating the palmitoylation sites caused an increase in PM EGFR levels and delayed trafficking to the lysosome. If this study is clarifying previous studies, the authors need to genetically inhibit the relevant DHHC enzyme and show decreased EGFR PM localization and use mass spectrometry to identify the relevant palmitoylation site(s) that are important for this mechanism.

The proposed mechanism for Arf6 shRNA inhibition of tumor growth is through reducing EGFR signaling. Both A549 and MDA-MB-231 have wild type EGFR status and both harbor activating mutations in Kras that are thought to be the main oncogenic driver. Neither cell line responds well to EGFR inhibitors. It seems that the mechanism of tumor growth inhibition may not be specific to EGFR signaling and it should be determined if Arf6 is simply an essential gene since it is required

for normal function of multiple organelles. The Arf6 knock out mouse is embryonic lethal which suggests Arf6 is important in many cell types. Therefore, the work does not support the conclusions and claims and I can't recommend this for publication.

Point by point critique

- 1) It is unexpected that mutating any cysteine on the intracellular domain blocked EGFR plasma membrane localization, and this raises the question of why Fig S1B does not include localization of the tagged wild-type receptor to compare with the localization of single cysteine mutants. That would help ensure there isn't something defective in the construct. Do these mutations still respond to EGF? Truong, et al., *Cell Chem Biology* (2016) expressed C797S/A mutant in Hela cells and although they didn't show the cell localization, they did show the receptor in Hela cells responds to EGF indicating it is on the plasma membrane. Runkle et al. *Molecular Cell* (2016) showed FLAG tagged EGFR C1049A (aka C1025) and C1146A (aka C1122) still localize to the plasma membrane.
- 2) If mutating any of the cytosolic cysteine residues is sufficient to block PM localization (Fig S1B) why does the study use the 9-cysteine mutant throughout?
- 3) A plasma membrane marker is needed to visualize where the PM is in cells where EGFR is determined to be not at the PM. It is difficult to tell if the PM localization is blocked or is the cell flatter and the edge of the membrane is less obvious. A plasma membrane marker will also show if knocking down Arf6 or expressing 9C-EGFR is blocking all trafficking from the Golgi to the plasma membrane. What else does or doesn't traffic to the PM when Arf6 is knocked down.
- 4) In figure 2a Arf6 should be detected by western blotting. There is no way to know the band shown in the silver-stained gel is Arf6. The legend says mass spec was performed to identify Arf6, but there is no mass spec data showing this and the text doesn't say how Arf6 was identified in the gel.
- 5) Figure 2b needs a western blot confirming Arf6 is ablated in the Arf6^{-/-} cells.
- 6) It is also unexpected that the 9C-EGFR mutant prevents Arf6 from localizing to the Golgi apparatus (Fig 2d). It seems like Arf6 would have a larger role at the Golgi than to traffic EGFR. There should be multiple cells counted and a correlation coefficient determined to make these conclusions.
- 7) Data showing the effect of the myristoylated peptide on EGFR trafficking, levels and signaling in cells is needed. In Figure 7a it isn't clear if EGFR is trapped in the Golgi or if it is localized in endosomes.
- 8) Did the authors try to measure EGFR and EGFR signaling in the peptide treated tumors compared to the vehicle. This would help confirm the effect on tumor growth is through their proposed mechanism.

Reviewer #3 (Remarks to the Author):

Manuscript by Guo et al entitled "Targeting EGFR-dependent tumors by disrupting an Arf6-mediated sorting system" has identified the role of EGFR palmitoylation and Arf6 myristoylation as critical factors in guiding EGFR budding from Golgi complex and its transport to the plasma membrane. This manuscript has the potential of being high impact and the manuscript and may be suitable for publication on the *Nat Communications* after revision. We have several comments that need to be addressed to strengthen the manuscript.

1. Authors have created cysteine to serine mutations in the kinase domain and autophosphorylation domain of the EGFR cytoplasmic domain. Simultaneous mutations of all cysteine residues to serine may provide the artificial serine phosphorylation sites in the cytoplasmic domain. Have author done any work to rule out this? In creating cysteine to serine mutations, this is of particular importance if these residues are in the sorting motifs of the cytoplasmic domain of the EGFR (e.g. C1146).
2. Authors have indicated that multiple cysteine residues are palmitoylated and are responsible for EGFR budding from Golgi and its transport to the plasma membrane. Also, suggested that more than one palmitoylation site is required for directing the EGFR to the plasma membrane. However, authors have failed to demonstrate the specific cysteine residues that are key for EGFR

palmitoylation and EGFR trafficking to the plasma membrane. Some cysteine residue mutant forms such as C775 and C781 seem palmitoylated comparable to wild type EGFR. If multiple residues are involved in the EGFR palmitoylation and play a role in EGFR trafficking, then, authors could have seen incremental defects in EGFR palmitoylation and its importation to the plasma membrane in alignment with number of cysteine residues mutated. Furthermore, mutant forms such as C775 and C781 appear similarly palmitoylated as wild type EGFR but show a defect in plasma membrane transport indicating that the defective EGFR transport to the plasma membrane is beyond palmitoylation at least in the context of the C775 and C781 mutations. Authors need to provide a convincing explanation for this.

3a. The EGFR and Arf6 association is based on coimmunoprecipitation assay, not *in vitro* binding of the purified proteins or a similar approach, authors should refrain from using the term "binding affinity" of interaction at page 8 first paragraph.

3b. The colocalization of EGFR and Arf6 does not indicate association; the authors should also use a proximity ligation assay which can be semi-quantitative.

4. The Arfs are myristoylated, as noted in the Supplementary Fig. 3a. Gly 2 appears conserved in all Arfs at the myristylation site. In defining the specificity of the Arf6 among other Arfs in association with EGFR, authors focused on N-terminus GKVL residues. However, authors concluded that none of these amino acid residues are involved in direct interaction with EGFR, but myristylation is a critical factor in EGFR association and in regulation of EGFR transport to the plasma membrane. Thus, if Gly 2 and its myristoylation is the essential factor for EGFR association and its transport to the plasma membrane, why do other Arfs that are also myristoylated lack their ability to mediate EGFR transport to the plasma membrane. The authors need to address this key point.

5. EGFR and Arf6 association spatially takes place at the Golgi complex before being directed to the plasma membrane, and the exocyst complex is recruited by Arf6 cysteine 3. Specifically, EGFR and Arf6 association is mediated indirectly by the palmitoyl moiety attached with EGFR and the myristoyl moiety attached with Arf6. However, others have demonstrated that the palmitoylation of EGFR is required for its association with myristoylated Arf6, not lipid-lipid interaction of palmitoyl and myristoyl moieties as claimed by authors in this manuscript. This needs to be addressed.

6. In an attempt to identify the protein complex associated with the EGFR and Arf6 complex at the Golgi complex, authors have used the microtubule depolymerizing agent nocodazole. Presumably authors used nocodazole to disrupt microtubules and vesicle trafficking and enrich and/or entrap the EGFR/Arf6 complex and its interacting partners at the Golgi complex. Authors need to mention this important point in the result section in the text.

7. Does GDP- vs. GTP-bound Arf6 complex dictate or regulate the exocyst complex recruitment to the EGFR/Arf6 complex?

8. Authors have heavily used overexpression strategies. Have authors recapitulated results displayed in Figure 1 using EGFR overexpressing cell lines, such as MDA-MB-231 and A549 cells?

9. The role of EGFR palmitoylation and its transport to the plasma membrane and its regulation of downstream signaling, such as Erk1/2 and/or PI3K/Akt signaling, would have been more compelling.

10. The authors have used cell lysates to suggest an interaction in Figure 4e. Have the authors attempted to demonstrate the direct interaction between Arf6 and EXOC2 using proteins purified from bacteria or eukaryotic expression approaches?

Reviewers' comments:

Reviewer #1 (Remarks to the Author):

The findings reported by Guo et al are overall interesting. The concept of post-transductional modifications such as palmitoylation and myristoylation affecting trafficking and receptor function is interesting. However, from reading the paper, several issues can be raised to challenge the validity of the conclusions presented by the authors.

Fig 1: 2-BP treatment will affect the function of broad range of proteins. The use of such inhibitors does not prove the authors point. The function of the 9CS EGFR mutant has not been thoroughly characterized. Could misfolding of the receptor explain its retention in the Golgi?

Response: Thank you for raising the concern. To strength our point that palmitoylation is required for the plasma membrane localization of EGFR, we provided more evidences as the followings.

First, we performed an shRNA screening of Golgi-localized DHHCs in MDA-MB-231 cells, which expressed endogenous EGFR, and found that knockdown of DHHC13 disrupted EGFR palmitoylation, plasma membrane localization and signaling (Fig.1e-g; Supplementary Fig. 1c,d). And we further confirmed these results in A549 cells, another cell line with endogenous expression of EGFR (Supplementary Fig. 1f-g).

We then went on to identify the palmitoylation sites of EGFR by DHHC13. By mutating 8 of the 9 cytosolic Cys to Ser, i.e. keeping only one cytosolic Cys, we showed that C775, C781 and C797 of EGFR are palmitoylated by DHHC13 (Fig. 1h; Supplementary Fig.1h). And mutation of these 3 Cys (C775SC781SC797S) failed to target EGFR to plasma membrane and trapped EGFR in Golgi (Fig.1i). We therefore used the 3CS mutant for the rest of the study.

With these data, we hope we can convince you that palmitoylation of EGFR by DHHC13 is required for EGFR PM localization.

Fig 2: ARF6 is the isoform found preponderantly at the plasma membrane. Why here, is it mainly intracellular? KD of ARF6 has been shown to regulate endocytosis of plasma membrane receptors. Here, it is unclear where ARF6 precisely acts. The use of overexpressed Flag-ARF6 (modified GTPase) might not represent the function of endogenously expressed ARF since modifications of the GTPase is known to alter function. The amount of overexpression might also impair localization and explain finding in 2C.

Response: We agree that ARF6 is predominantly localized on the plasma membrane. Actually, at the 1 hr time point of the RUSH system in Fig. 2e (Fig. 2c in the previous version), Fig. 3a and Fig. 3b, we show that ARF6 is mainly localized on the plasma membrane, but also exhibits intracellular localization. At the 0 timepoint of the RUSH system in Fig. 2e, ARF6 indeed showed relatively more Golgi localization at the retention state, which was recruited by palmitoylated EGFR as supported by other evidences (Fig. 2f,g).

In Fig. 2e-g, the experiments were done in *ARF6*^{-/-} cells. While loss of ARF6 disrupted

the plasma membrane localization of EGFR (Fig. 2d), re-introduction of ARF6-Flag restored the plasma membrane localization of EGFR at the 1 hr timepoint of the RUSH system (Fig. 2e). Similarly, in Fig. 3a and 3b, ARF6-HA restored the plasma membrane localization of EGFR in *ARF6*^{-/-} cells. These results indicate that adding a C-terminal Flag or HA tag does not affect the function of ARF6.

We also examined the expression level of ARF6-Flag in *ARF6*^{-/-} cells and found that it was very close to the endogenous level of ARF6 in the WT cells (Supplementary Fig. 2c).

Fig 3: ARF5 and ARF6 chimeras are used, but no thorough characterization of the roles of all ARF isoforms either alone or in combination have been done.

Response: Thank you for the suggestion. We also examined the other forms of ARFs (ARF1, ARF3 and ARF4), and found that none of them would rescue the plasma localization of EGFR in *ARF6*^{-/-} HeLa cells. The data about the other forms of ARFs are now included in Supplementary Fig. 3b.

Fig 5: The use of ARF mutants has to be done acknowledging the weaknesses these have. The consensus about the use of the constitutively active ARF6QL mutant is that inhibiting cycling might results in many cases as dominant negative effects. The only expression of this mutant alters cellular morphology. A fast cycling mutant needs to be tested.

Response: Thanks for your advice. We generated a fast cycling ARF6 mutant T157A according to a previous article (*J Biol Chem*, 2002,277(43):40185-8.), and reintroduced it into *ARF6*^{-/-} cells. As shown in Supplementary Fig. 4, the T157A mutant of ARF6-GFP also rescued the EGFR PM localization, indicating that the GTP-bound form of ARF6 is required for its function in targeting EGFR to plasma membrane.

Fig 6: The effect of ARF6 on EGFR function in TNBC cells is not novel. The data presented here do not give further insights.

Response: Thank for raising the point. Indeed, we noticed that several papers from Dr. Sabe's group reported the critical role of ARF6 in the TNBC invasion and metastasis. Especially in the article published in *Nat Cell Biol* (2008,10(1):85-92. doi: 10.1038/ncb1672), the authors show that GEP100, an ARF6 GEF, binds Tyr1068/1086-phosphorylated EGFR to activate ARF6 to induce TNBC invasion and metastasis. Although both the NCB paper and ours highlighted the critical roles of EGFR and ARF6 in TNBC, the emphases were different. The NCB paper mainly focused on the role of EGFR in activating ARF6, using TNBC invasion and metastasis as the readout, whereas our study weighs on the role of ARF6 in regulating EGFR function, using tumor growth as a readout. Also, the tumor model presented in Fig. 6 is a NSCLC A549 tumor model, not TNBC. With this, we hope the reviewer can agree with us that the data in Fig. 6 still show the novelty, especially about the regulation of EGFR by ARF6.

There is also an article published in *Nature Communications* (2017, DOI: 10.1038/ncomms14688) showed that EGFR works through ARF6 to stimulate tumor oncogenic Ras tumor overgrowth. This is actually consistent with our findings made in MDA-MB-231 and A549 cells, both of which bear KRas mutations. To better explain the data, we included the major observations of the two papers in the "Discussion" section as

followings.

“Notably, EGFR has also been shown to regulate the activation of ARF6. In breast cancers, GEF100, another GEF protein of ARF6, has been shown to bind Tyr1068/1086-phosphorylated EGFR to activate ARF6 to induce TNBC invasion and metastasis³². EGFR also activates ARF6 to stimulate oncogenic Ras tumor overgrowth by regulating Hh signaling³³. In fact, MDA-MB-231 and A549 cells used in our experiments also bear KRas mutations^{16, 17}, and we show that knockdown of either EGFR or ARF6 inhibits tumor growth. Combining the previous observations with our results, we propose that ARF6 and EGFR might regulate the activity of each other to promote tumor progression by forming a positive feedback loop. On the one hand, ARF6 controls EGFR signaling by transporting EGFR to PM; on the other hand, EGFR on PM binds its ligands to activate ARF6.”

Reviewer #2 (Remarks to the Author):

The manuscript by Huiling Guo et al. examines the requirement of the small GTPase Arf6 in trafficking EGFR from the Golgi apparatus to the plasma membrane. Disruption of this mechanism by shRNA targeting Arf6 blocks EGFR PM localization, EGFR signaling and tumor growth in xenograft assays. A myristoylated peptide from Arf6 also blocks tumor growth in mice. The authors propose palmitoylation of EGFR is required to be recognized by Arf6 which recruits the exocyst complex to mediate trafficking to the plasma membrane. While it has been shown that the exocyst complex is involved in EGFR trafficking to the PM in stem cells and in kidney tissues, the involvement of Arf6 appears to be a new finding.

General Critique

My main concern with this manuscript is that the studies with EGFR all use an expression construct that has 9-cysteine residues mutated in the cytosolic domain. Has it been shown that EGFR still folds properly when 6 of the cysteine residues in the kinase domain are mutated or could it be unfolded/partially folded and trapped in the ER/Golgi compartments? If the protein is unfolded, it probably won't be palmitoylated. Most of the experiments in this study rely on 2-BP which is not a specific palmitoyl transferase inhibitor. It is a convenient first step towards showing a protein is palmitoylated, but the effect of 2BP on complex cellular processes cannot be relied on and needs verification by genetically inhibiting the relevant transferase. Runkle et al. *Molecular Cell* (2016) showed that EGFR is palmitoylated by DHHC20 and identified two palmitoylation sites in the unstructured C-terminal tail by mass spectrometry. Knocking down DHHC20 or mutating the palmitoylation sites caused an increase in PM EGFR levels and delayed trafficking to the lysosome. If this study is clarifying previous studies, the authors need to genetically inhibit the relevant DHHC enzyme and show decreased EGFR PM localization and use mass spectrometry to identify the relevant palmitoylation site(s) that are important for this mechanism.

Response: Thank you for your suggestions. To strength our point that palmitoylation is required for the plasma membrane localization of EGFR, we provided more evidences as the followings.

To confirm the results obtained from 2-BP treatment and the 9CS mutant, we performed an shRNA screening of Golgi-localized DHHCs in MDA-MB-231 cells and identified DHHC13 is required for EGFR PM localization (Supplementary Fig. 1c,d). Knockdown of DHHC13 decreased the palmitoylation of EGFR, disrupted its plasma membrane localization, and inhibited EGF-induced EGFR signaling (Fig.1e-g). And we further confirmed these results in A549 cells (Supplementary Fig. 1f-g).

We then went on to identify the palmitoylation sites of EGFR by DHHC13. By mutating 8 of the 9 cytosolic Cys to Ser, i.e. keeping only one cytosolic Cys, we showed that C775, C781 and C797 of EGFR are palmitoylated by DHHC13 (Fig. 1h; Supplementary Fig.1h). And similar to the 9CS mutant (Fig. 1c), the 3CS mutant (C775SC781SC797S) was also trapped in Golgi and failed to reach plasma membrane (Fig.1i). We therefore used the 3CS mutant for the rest of the study.

With these data, we hope we can convince you that palmitoylation of EGFR by DHHC13 is required for EGFR PM localization.

The proposed mechanism for Arf6 shRNA inhibition of tumor growth is through reducing EGFR signaling. Both A549 and MDA-MB-231 have wild type EGFR status and both harbor activating mutations in Kras that are thought to be the main oncogenic driver. Neither cell line responds well to EGFR inhibitors. It seems that the mechanism of tumor growth inhibition may not be specific to EGFR signaling and it should be determined if Arf6 is simply an essential gene since it is required for normal function of multiple organelles. The Arf6 knock out mouse is embryonic lethal which suggests Arf6 is important in many cell types.

Response: Thank you for your question. Although both A549 and MDA-MB-231 cells harboring KRas mutations, we knocked down EGFR and found it significantly inhibited cell growth of both cell lines (Fig. 6f; Supplementary Fig. 6f), similar to the effect of knocking down ARF6. Double knockdown of ARF6 and EGFR did not further decrease cell growth (Fig. 6f; Supplementary Fig. 6f).

In consistence with our data, a previous paper (Nature Communications, 2017, DOI: 10.1038/ncomms14688) showed that EGFR works through ARF6 to stimulate tumor oncogenic Ras tumor growth by regulating Hh signaling. The authors showed that knockout of EGFR or ARF6 significantly suppressed *Ras*^{V12} tumor growth. Although different cell lines were used and the mechanisms were different, both manuscripts support that EGFR and ARF6 play a critical role in the tumor growth of oncogenic Ras tumors. To help to explain the results, we added the following paragraph in the “Discussion” section.

“EGFR also activates ARF6 to stimulate oncogenic Ras tumor overgrowth by regulating Hh signaling³³. In fact, MDA-MB-231 and A549 cells used in our experiments also bear KRas mutations^{16, 17}, and we show that knockdown of either EGFR or ARF6 inhibits tumor growth.”

Point by point critique

1) It is unexpected that mutating any cysteine on the intracellular domain blocked EGFR plasma membrane localization, and this raises the question of why Fig S1B does not include localization of the tagged wild-type receptor to compare with the localization of single cysteine mutants. That would help ensure there isn't something defective in the construct. Do these mutations still respond to EGF? Truong, et al., Cell Chem Biology (2016) expressed C797S/A mutant in Hela cells and although they didn't show the cell localization, they did show the receptor in Hela cells responds to EGF indicating it is on the plasma membrane. Runkle et al. Molecular Cell (2016) showed FLAG tagged EGFR C1049A (aka C1025) and C1146A (aka C1122) still localize to the plasma membrane.

Response: Thank you for your comments, and we apologize for the misunderstanding. All the EGFR mutants in Supplementary Fig.1b of the previous submission and Supplementary Fig.1h in the revised version contained only one Cys in the cytosolic domain and the other 8 Cys residues were all mutated to Ser. To avoid misunderstanding, we labeled the mutants as 8CS with the remaining Cys on the top of each panel in the revised manuscript. The results showed in the previous Supplementary Fig.1b revealed that a single Cys is not sufficient for EGFR targeting to PM.

As mentioned earlier, in revised manuscript, we identified DHHC13 as the palmitoylating

enzyme of EGFR, and identified C775, C781 and C797 of EGFR as the palmitoylation sites of DHHC13 (Fig. 1e-g). And we went on to show that mutation of these three Cys disrupted the plasma membrane localization of EGFR (Fig. 1h,i). As these results provided a better explanation of how palmitoylation of EGFR controls its plasma membrane localization, we replace the previous Supplementary Fig. 1b with the current figures.

2) If mutating any of the cytosolic cysteine residues is sufficient to block PM localization (Fig S1B) why does the study use the 9-cysteine mutant throughout?

Response: Again, we apologize for the misunderstanding. All the EGFR mutants in Supplementary Fig. 1b of the previous submission contained only one Cys in the cytosolic domain and the other 8 Cys residues were all mutated to Ser. We have made changes to the labels of the mutants to avoid further misunderstanding.

3) A plasma membrane marker is needed to visualize where the PM is in cells where EGFR is determined to be not at the PM. It is difficult to tell if the PM localization is blocked or is the cell flatter and the edge of the membrane is less obvious. A plasma membrane marker will also show if knocking down Arf6 or expressing 9C-EGFR is blocking all trafficking from the Golgi to the plasma membrane. What else does or doesn't traffic to the PM when Arf6 is knocked down.

Response: Thank you for your suggestion. In the revised manuscript, we used FITC or Rhodamine-labeled phalloidin to stain F-actin to indicate plasma membrane. And the new results were consistent with the previous conclusions (Fig. 1 and Fig. 2).

In terms of "what else does or doesn't traffic to the PM when Arf6 is knocked down", in a separate study we performed SILAC in WT and ARF6^{-/-} cells aiming at finding out what other proteins would require ARF6 for plasma membrane localization. Our results showed that the plasma membrane content of some of the proteins were affected by ARF6, and some not. As shown below, HER2, a homolog of EGFR, was also palmitoylated and its plasma membrane localization was dependent on ARF6. However, the localization of another palmitoylated protein DSC2 was not dependent on ARF6, which could be due to different sorting pathway was used. We are still working on the detailed mechanisms. As we have demonstrated that ARF6 works through EGFR to control EGFR highly expressed tumor progression (Fig. 6f; Supplementary Fig. 5f), characterization of other substrates of ARF6 might be distracting the audience. We therefore asked not to include the SILAC data in the current manuscript.

Figure for the reviewer. Identification of proteins that rely on ARF6 for PM localization

a,b, WT and $ARF6^{-/-}$ HeLa cells were SILAC labeled and pooled for isolation of palmitoylated proteins on the PM. **b**, The ratio of each protein in WT versus $ARF6^{-/-}$ cells was plotted in the right panel. **c,e**, Palmitoylation analysis of HER2 and DSC2 by Acyl-RAC assay. **d,f**, HER2-Flag (**d**) or DSC2-Flag (**f**) were transfected into WT or $ARF6^{-/-}$ HeLa cells. Immunostaining was performed as in Fig. 1c.

4) In figure 2a Arf6 should be detected by western blotting. There is no way to know the band shown in the silver-stained gel is Arf6. The legend says mass spec was performed to identify Arf6, but there is no mass spec data showing this and the text doesn't say how Arf6 was identified in the gel.

Response: Thank you for your suggestion. To make the data more convincing, we included a spectrum of the peptide identified from ARF6 by the mass spec in Supplementary Fig. 2a.

5) Figure 2b needs a western blot confirming Arf6 is ablated in the $Arf6^{-/-}$ cells.

Response: Thank you. We have confirmed that ARF6 protein was ablated in the $Arf6^{-/-}$ cells, and these data were now included the data in Supplementary Fig. 2c.

6) It is also unexpected that the 9C-EGFR mutant prevents Arf6 from localizing to the Golgi apparatus (Fig 2d). It seems like Arf6 would have a larger role at the Golgi than to traffic EGFR. There should be multiple cells counted and a correlation coefficient determined to make these conclusions.

Response: Thank you for your suggestion. We reperfomed the experiment by co-expressing ARF6-GFP, Golgin-84-Streptavidin and WT or 3CS of EGFR-HA-SBP in *ARF6*^{-/-} cells. As shown in Supplementary Fig. 2d and 2e, the mean Manders' coefficient for ARF6 and WT EGFR is about 0.6, whereas it is 0.27 for ARF6 and the 3CS mutant. These data further confirmed that the palmitoylation of EGFR is required for its interaction with ARF6.

7) Data showing the effect of the myristoylated peptide on EGFR trafficking, levels and signaling in cells is needed. In Figure 7a it isn't clear if EGFR is trapped in the Golgi or if it is localized in endosomes.

Response: Thank you for your advice. We performed the experiments and found that treatment with Myr-GKVL-TAT dramatically decreased the protein levels and signaling of EGFR in both MDA-MB-231 and A549 cells (Fig. 7b; Supplementary Fig. 6b). Myristoylated peptide treatment decreased EGF-stimulated EGFR phosphorylation, downstream p-AKT and p-ERK. And we reperfomed the immunostaining and found that trapped EGFR is at least partially colocalized with Golgi marker (Fig.7a and Supplementary Fig.6a).

8) Did the authors try to measure EGFR and EGFR signaling in the peptide treated tumors compared to the vehicle. This would help confirm the effect on tumor growth is through their proposed mechanism.

Response: Thank you for your suggestion. We performed the experiment and found that Myr-GKVL-TAT treatment decreased EGF-stimulated EGFR phosphorylation, downstream phospho-AKT and phospho-ERK (Supplementary Fig. 6f,g).

Reviewer #3 (Remarks to the Author):

Manuscript by Guo et al entitled “Targeting EGFR-dependent tumors by disrupting an Arf6-mediated sorting system” has identified the role of EGFR palmitoylation and Arf6 myristoylation as critical factors in guiding EGFR budding from Golgi complex and its transport to the plasma membrane. This manuscript has the potential of being high impact and the manuscript and may be suitable for publication on the Nat Communications after revision. We have several comments that need to be addressed to strengthen the manuscript.

1. Authors have created cysteine to serine mutations in the kinase domain and autophosphorylation domain of the EGFR cytoplasmic domain. Simultaneous mutations of all cysteine residues to serine may provide the artificial serine phosphorylation sites in the cytoplasmic domain. Have author done any work to rule out this? In creating cysteine to serine mutations, this is of particular importance if these residues are in the sorting motifs of the cytoplasmic domain of the EGFR (e.g. C1146).

Response: Thank you for your comments. To rule out artificial effect of mutations, we screened the DHHC enzyme and identified DHHC13 as the palmitoylating enzyme required for EGFR palmitoylation, plasma membrane localization and signaling (Fig.1e-g; Supplementary Fig.1d-g). Furthermore, we identified C775, C781 and C797 as the major sites of DHHC13, and demonstrated that mutation of these sites disrupted the plasma membrane localization of EGFR (Fig. 1h,i; Supplementary Fig. 1h).

2. Authors have indicated that multiple cysteine residues are palmitoylated and are responsible for EGFR budding from Golgi and its transport to the plasma membrane. Also, suggested that more than one palmitoylation site is required for directing the EGFR to the plasma membrane. However, authors have failed to demonstrate the specific cysteine residues that are key for EGFR palmitoylation and EGFR trafficking to the plasma membrane. Some cysteine residue mutant forms such as C775 and C781 seems palmitoylated comparable to wild type EGFR. If multiple residues are involved in the EGFR palmitoylation and play role in EGFR trafficking, then, authors could have seen incremental defects in EGFR palmitoylation and its importation to the plasma membrane in alignment with number of cysteine residues mutated. Furthermore, mutant forms such as C775 and C781 appear similarly palmitoylated as wild type EGFR but show a defect in plasma membrane transport indicating that the defective EGFR transport to the plasma membrane is beyond palmitoylation at least in the context of the C775 and C781 mutations. Authors need to provide convincing explanation for this.

Response: Thank you for your comments. As mentioned above, we carried out more experiments and C775, C781 and C797 as the major palmitoylation sites that is required for the plasma membrane localization of EGFR (Fig. 1h,i; Supplementary Fig. 1h).

3a. The EGFR and Arf6 association are based on coimmunoprecipitation assay, not vitro binding of the purified proteins or a similar approach, authors should refrain from using

term “binding affinity” of interaction at page 8 first paragraph.

Response: Thank you for your suggestion. We have changed the description “binding affinity” to “interaction”.

3b. The colocalization of EGFR and Arf6 does not indicate association the authors should also use a proximity ligation assay which can be semi – quantitative.

Response: Thank you for your advice. We purchased the proximity ligation assay kit from Merck (Sigma) and performed the experiment. As shown in Fig. 2h and 2i, wild types of ARF6 and EGFR were spatially close enough to generate the fluorescent signal, which was much higher than that from the 3CS mutant of EGFR and G2A mutant of ARF6.

4. The Arfs are myristoylated, as noted in the Supplementary Fig. 3a Gly 2 appears conserved in all Arfs at the myristylation site. In defining the specificity of the Arf6 among other Arfs in association with EGFR, authors focused on N-terminus GKVL residues. However, authors concluded that none of these amino acid residues are involved in direct interaction with EGFR, but myristylation as critical factor in EGFR association and in regulation of EGFR transport to the plasma membrane. Thus, if Gly 2 and its myristoylation is the essential factor for EGFR association and its transport to the plasma membrane, why do other Arfs that are also myristoylated lack their ability to mediate EGFR transport to the plasma membrane. The authors need to address this key point.

Response: Thank you for your comments. As you pointed out, we show that the myristoylation of Gly2 in ARF6 is required for its binding with EGFR. Indeed, all the other forms of ARFs have Gly2; however, when introduced into *ARF6*^{-/-} cells, none of them could rescue the plasma membrane localization of EGFR, though they all show some colocalization with EGFR in Golgi (Fig. 3a; Supplementary Fig. 3b). These data indicate that other residues in the N-terminus of ARF6 is required for its function. We went on to show that Lys3, which is unique in ARF6, is required for EGFR targeting to the plasma membrane (Fig. 3b,c). Lys3 is recognized by EXOC2 of the Exocyst complex to facilitate EGFR sorting to the plasma membrane (Fig. 4). Based on these data, Gly2 is required for binding of EGFR, whereas Lys3 is recognized by the Exocyst complex to facilitate sorting of EGFR.

5. EGFR and Arf6 association spatially takes place at the Golgi complex before being directed to the plasma membrane, and exocyst complex is recruited by Arf6 cysteine 3. Specifically, EGFR and Arf6 association is mediated indirectly by palmitoyl moiety attached with EGFR and myristoyl moiety attached with Arf6. However, others have demonstrated the palmitoylation of EGFR is required for its association with myristoylated Arf6, not lipid-lipid interaction of palmitoyl and myristoyl moieties as claimed by authors in this manuscript. This needs to be addressed.

Response: Thank you for your comments. In a previous article published in *Nat Cell Biol* (2008,10(1):85-92. doi: 10.1038/ncb1672), the authors showed that GEP100, an ARF6 GEF, binds Tyr1068/1086-phosphorylated EGFR to activate ARF6. It seems like that GEP100 might mediate the interaction between EGFR and ARF6, but the authors did not provide direct evidence. In the revised manuscript, we provided several lines of evidences

to show that palmitoyl moiety of EGFR and the myristoyl moiety of G2A are required for their interaction. First, by immunoprecipitation, lack of either palmitoylation or myristoylation decreased the interaction between EGFR and ARF6 (Fig. 2c, j, k). Second, by immunofluorescence analysis in the RUSH system, lack of either of the lipid modifications largely decreased the colocalization of EGFR and ARF6 in the Golgi (Fig. 2f,g). Third, by proximity ligation assay, lack of the lipid modifications dramatically decreased the interaction between EGFR and ARF6 (Fig. 2h,i). With these data, we hope we can convince you that the EGFR interacts with ARF6 through lipid-lipid interaction.

6. In an attempt to identify the protein complex associated with EGFR and Arf6 complex at Golgi complex, authors have used microtubule depolymerizing agent nocodazole. Presumably authors used nocodazole to disrupt microtubules and vesicles trafficking and enrich and/or entrap the EGFR/Arf6 complex and its interacting partners at the Golgi complex. Authors need to mention this important point in the result section in the text.

Response: Thank you for your suggestion. We have added the description in the result of Fig. 4a. "Nocodazole was used to disrupt microtubules and vesicle trafficking to enrich the EGFR/Arf6 complex and its interacting partners at the Golgi complex."

7. Does GDP- vs. GTP-bound Arf6 complex dictate or regulate the exocyst complex recruitment to EGFR/Arf6 complex?

Response: Thank you for your question. We co-expressed WT, T27N or Q67L of ARF6 with Flag-EXOC2 in HEK-293T cells and performed coimmunoprecipitations. WT ARF6 and Q67L mutant, which mimics GTP-bound form, exhibited similar interactions with EXOC2, but the T27N mutant, which mimics GDP-bound form, showed weaker interaction (Fig. 5b). These data indicate that activation of ARF6 enhances its association with the exocyst complex.

8. Authors have heavily used overexpression strategies. Have author recapitulated results displayed in Figure 1 using EGFR overexpressing cell lines, such as MDA-MB-231 and A549 cells?

Response: Thank you for your suggestion. We performed the experiments in MDA-MB-231 and A549 cells, and recapitulated the results. The new data were presented in Fig. 1a, b, e-g and Supplementary Fig. 1c-g.

9. The role of EGFR palmitoylation and its transport to plasma membrane and its regulation of downstream signaling, such as Erk1/2 and/or PI3K/Akt signaling would have been more compelling.

Response: Thank you for your suggestion. In the revised manuscript, we added the data about p-ERK and p-AKT in the experiments to further strengthen the regulation of EGFR localization and signaling by its palmitoylation. These data were included in Fig. 1g, Fig. 6b, Fig. 7b, Supplementary Fig. 1g, and Supplementary Fig, 6b,f,g. All the data support that EGFR palmitoylation is required for its plasma membrane localization and downstream signaling.

10. The author have used cell lysates to suggest an interaction in Figure 4e. Have the authors attempted to demonstrate the direct interaction between Arf6 and EXOC2 using proteins purified from bacteria or eukaryotic expression approaches?

Response: Thank you. The experiment in Fig. 4e (Fig. 4f in the revised version) was actually done with purified proteins. WT and mutants of ARF6-Flag were expressed and purified from HEK-293T cells, and GST-EXOC2 were expressed and purified from *E.coli*. We have included the information in the figure legends.

REVIEWER COMMENTS

Reviewer #1 (Remarks to the Author):

The authors have addressed very seriously the concerns of the 3 different reviewers. Their added work confirm their hypothesis and greatly strengthen their paper.

Reviewer #2 (Remarks to the Author):

The revised manuscript by Huiling Guo et al. examines the requirement of the small GTPase Arf6 in trafficking EGFR from the Golgi apparatus to the plasma membrane. Disruption of this mechanism by shRNA targeting Arf6 blocks EGFR PM localization, EGFR signaling and tumor growth in xenograft assays. A myristoylated peptide from Arf6 also blocks tumor growth in mice. The authors propose palmitoylation of EGFR is required to be recognized by Arf6 which recruits the exocyst complex to mediate trafficking to the plasma membrane.

The authors have added significant additional data that have strengthened the manuscript overall addressing many of my concerns.

My main concern remains that it is unclear if it is loss of palmitoylation of the cysteine residues that results in loss of trafficking of EGFR to the plasma membrane and not structural misfolding or altered activity caused by mutating 3 cysteine residues in the kinase domain that causes the phenotype.

General Critique

The authors have reduced the number of mutated cysteine residues in the kinase domain down to 3 which is an improvement over the 9 mutations in the original manuscript. It is reassuring that knock down of DHHHC13 also causes accumulation of EGFR in the Golgi and this does strengthen their conclusion. Showing localization of DHHHC13 to the Golgi in their system would further strengthen these results. Showing DSC2 or another protein traffics normally to the PM in cells expressing shZDHHHC13 to show normal trafficking of some proteins in that context is needed.

Because the mutations cause a loss of function and not a gain of function effect, it is still difficult to conclude with certainty it is caused by not being palmitoylated and not caused by structural misfolding or aggregation of the 3CS mutant EGFR even though shZDHHHC13 phenocopies the 3C mutant.

It is essential to show that the kinase domain is still folding correctly. Can the authors express just the kinase domain with the mutated cysteine residues and show it is still folded and soluble by running it on a size exclusion column or show it still binds ATP? Anything to show the protein is still folded and functional.

Showing a decrease in palmitoylation when mutating a cysteine only shows that cysteine is required for palmitoylation it doesn't mean that cysteine is palmitoylated. Mass spectrometry is the way to show a specific cysteine is palmitoylated. Concluding the kinase domain is palmitoylated needs to be rigorously demonstrated.

Reviewer #3 (Remarks to the Author):

The authors have worked to address our comments

Reviewer #2 (Remarks to the Author):

The revised manuscript by Huling Guo et al. examines the requirement of the small GTPase Arf6 in trafficking EGFR from the Golgi apparatus to the plasma membrane. Disruption of this mechanism by shRNA targeting Arf6 blocks EGFR PM localization, EGFR signaling and tumor growth in xenograft assays. A myristoylated peptide from Arf6 also blocks tumor growth in mice. The authors propose palmitoylation of EGFR is required to be recognized by Arf6 which recruits the exocyst complex to mediate trafficking to the plasma membrane.

The authors have added significant additional data that have strengthened the manuscript overall addressing many of my concerns.

My main concern remains that it is unclear if it is loss of palmitoylation of the cysteine residues that results in loss of trafficking of EGFR to the plasma membrane and not structural misfolding or altered activity caused by mutating 3 cysteine residues in the kinase domain that causes the phenotype.

General Critique

1. The authors have reduced the number of mutated cysteine residues in the kinase domain down to 3 which is an improvement over the 9 mutations in the original manuscript. It is reassuring that knock down of DHHC13 also causes accumulation of EGFR in the Golgi and this does strengthen their conclusion. Showing localization of DHHC13 to the Golgi in their system would further strengthen these results. Showing DSC2 or another protein traffics normally to the PM in cells expressing shZDHHC13 to show normal trafficking of some proteins in that context is needed.

Response: Thank you for your suggestions. First, we examined in the subcellular localization of DHHC13 and found that it was indeed localized in Golgi (Supplementary Fig. 1f). Second, we found that DSC2 trafficked normally to the PM in both control and DHHC13 knockdown cells (supplementary Fig.1h). With these data, we hope that we can convince you DHHC13 is the palmitoylating enzyme of EGFR.

2. Because the mutations cause a loss of function and not a gain of function effect, it is still difficult to conclude with certainty it is caused by not being palmitoylated and not caused by structural misfolding or aggregation of the 3CS mutant EGFR even though shZDHHC13 phenocopies the 3C mutant.

It is essential to show that the kinase domain is still folding correctly. Can the authors express just the kinase domain with the mutated cysteine residues and show it is still folded and soluble by running it on a size exclusion column or show it still binds ATP? Anything to show the protein is still folded and functional.

Response: Thank you for your suggestion. We generated the WT or 3CS mutant version of the kinase domain (696-960aa) of EGFR, expressed them in HEK-293T cells, isolated the cytosolic proteins and subjected to gel filtration. As shown in Supplementary Fig. 2c, the 3CS mutant behaved very similar to the WT version of the kinase domain, indicating that mutations in the 3 Cys did not affect protein folding of EGFR.

3. Showing a decrease in palmitoylation when mutating a cysteine only shows that cysteine is required for palmitoylation it doesn't mean that cysteine is palmitoylated. Mass spectrometry is the way to show a specific cysteine is palmitoylated. Concluding the kinase domain is palmitoylated needs to be rigorously demonstrated.

Response: Thank you for your suggestion. We have performed mass spectrometry (MS) to identify palmitoylation of cysteines. As shown in supplementary Fig. 2a, 8 of the 9 cytosolic of cysteines, except Cys781, were detected to be palmitoylated by MS. For Cys781, probably due to the digestion process during sample preparation, we could only detect very few peptides containing Cys781. And we could not rule out that Cys781 might also be palmitoylated.

To further confirm the results, we generated EGFR mutants that kept only one of the 9 cytosolic Cys and detected their palmitoylation by Acyl-RAC assay. The results in Supplementary Fig. 2b revealed that all of the 9 Cys were palmitoylated, and that Cys775, Cys781 and Cys797 were palmitoylated by DHHC13.

REVIEWERS' COMMENTS

Reviewer #2 (Remarks to the Author):

The authors have included mass spectrometry confirming the kinase domain of EGFR is palmitoylated at the proposed cysteine residues.

They also include gel filtration data showing the kinase domain with the 3 cysteine residues mutated to serine elutes similar to wild type kinase domain which is consistent with the mutant folding properly.

I have no further concerns and I am satisfied with the rigor of the study and the conclusions as reported by the authors.

REVIEWERS' COMMENTS

Reviewer #2 (Remarks to the Author):

The authors have included mass spectrometry confirming the kinase domain of EGFR is palmitoylated at the proposed cysteine residues.

They also include gel filtration data showing the kinase domain with the 3 cysteine residues mutated to serine elutes similar to wild type kinase domain which is consistent with the mutant folding properly.

I have no further concerns and I am satisfied with the rigor of the study and the conclusions as reported by the authors.

Response: Thank you for your positive comments.